# Aurora B switches relative strength of kinetochore–microtubule attachment modes for error correction

Harinath Doodhi, Taciana Kasciukovic, Lesley Clayton, and Tomoyuki U. Tanaka

To establish chromosome biorientation, aberrant kinetochore–microtubule interaction must be resolved (error correction) by Aurora B kinase. Aurora B differentially regulates kinetochore attachment to the microtubule plus end and its lateral side (end-on and lateral attachment, respectively). However, it is still unclear how kinetochore–microtubule interactions are exchanged during error correction. Here, we reconstituted the budding yeast kinetochore–microtubule interface in vitro by attaching the Ndc80 complexes to nanobeads. These Ndc80C nanobeads recapitulated in vitro the lateral and end-on attachments of authentic kinetochores on dynamic microtubules loaded with the Dam1 complex. This in vitro assay enabled the direct comparison of lateral and end-on attachment strength and showed that Dam1 phosphorylation by Aurora B makes the end-on attachment weaker than the lateral attachment. Similar reconstitutions with purified kinetochore particles were used for comparison. We suggest the Dam1 phosphorylation weakens interaction with the Ndc80 complex, disrupts the end-on attachment, and promotes the exchange to a new lateral attachment, leading to error correction.

## Introduction

For accurate and successful chromosome segregation, kinetochores must interact properly with spindle microtubules (MTs; Tanaka, 2010). The kinetochore initially interacts with the lateral side of a single MT (lateral attachment) and then becomes tethered at the MT plus end (end-on attachment) as the MT shrinks (Rieder and Alexander, 1990; Tanaka et al., 2007; Tanaka et al., 2005). Subsequently, sister kinetochores form end-on attachments to MTs extending from opposite spindle poles, establishing chromosome biorientation. If an aberrant kinetochore–MT attachment is formed, then it must be resolved (error correction) by Aurora B kinase (Ipl1 in budding yeast), which phosphorylates kinetochore components and disrupts the end-on attachment (Hauf et al., 2003; Lampson et al., 2004; Tanaka et al., 2002). In budding yeast, the Dam1 complex (Dam1C) is the most important Aurora B substrate for error correction (Cheeseman et al., 2002), and phosphorylation of Ndc80 N terminus also contributes to this process (Akiyoshi et al., 2009).

We previously showed that the end-on attachment is weakened by the action of Aurora B, but the lateral attachment is impervious to Aurora B regulation (i.e., the end-on and lateral attachments are differentially regulated; Kalantzaki et al., 2015). This led us to propose the model that during error correction, an end-on attachment is disrupted by the action of Aurora B (Fig. 1 A, steps 1 and 2) and subsequently replaced by lateral attachment to a different MT (steps 3 and 4); the lateral attachment is then converted to end-on attachment, and, if this results in aberrant attachment, it must be resolved again by Aurora B (step 1), but if biorientation is formed, then tension across sister kinetochores stabilizes end-on attachment (step 5; Kalantzaki et al., 2015). Thus, the model suggests that differential regulation of end-on and lateral attachments promotes the exchange of kinetochore–MT interactions for error correction.

A number of crucial questions still remain regarding the exchange of kinetochore–MT interactions during error correction. For example, although studies in budding yeast cells suggested the differential regulation of end-on and lateral attachments by Aurora B (Kalantzaki et al., 2015), this has not yet been directly tested during the exchange of kinetochore–MT interactions, as it is difficult to visualize in yeast cells. For exchange to occur efficiently, an end-on attachment to one MT must be disrupted and replaced by a lateral attachment to another MT. It is unknown how the relative strengths of lateral and end-on attachments are regulated by phosphorylation of kinetochore components by Aurora B kinase. Furthermore, although the Ndc80 complex (Ndc80C) and Dam1C are major outer kinetochore components comprising the kinetochore–MT interface (Jenni and Harrison, 2018; Kalantzaki et al., 2015;

......................................................................................................................................................................................................

Centre for Gene Regulation and Expression, School of Life Sciences, University of Dundee, Dundee, UK.

Correspondence to Tomoyuki U. Tanaka: t.tanaka@dundee.ac.uk; Harinath Doodhi: h.doodhi@dundee.ac.uk.

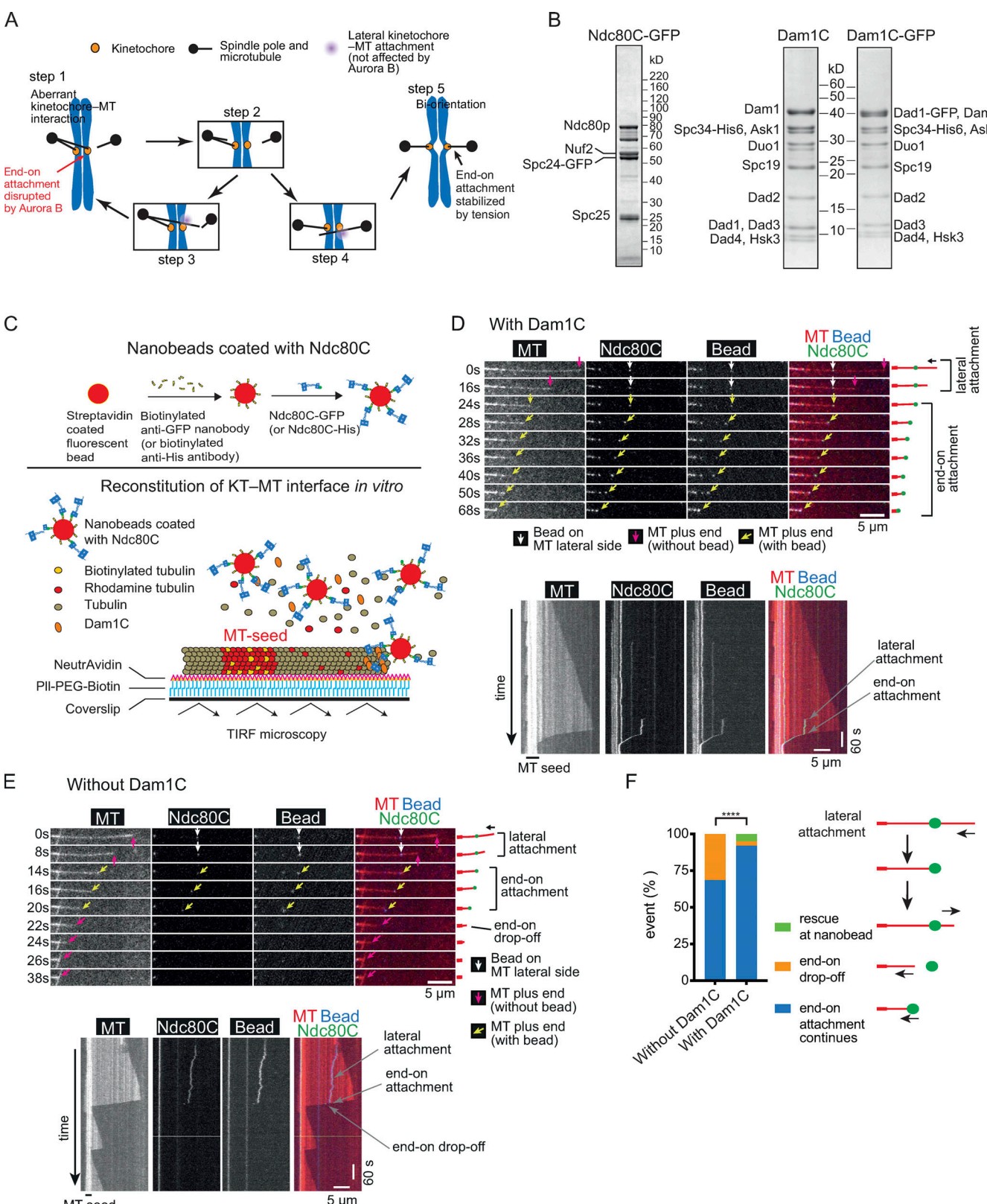

Figure 1. **Behavior of Ndc80C nanobeads in vitro on dynamic MTs loaded with Dam1C. (A)** Diagram shows the model of an error-correction process proposed previously (Kalantzaki et al., 2015). Each step is explained in the text. **(B)** Coomassie Blue–stained gels showing purified Ndc80C-GFP (with GFP at the C terminus of Spc24), Dam1C, and Dam1C-GFP (with GFP at the C terminus of Dad1). **(C)** Diagram shows that Ndc80C-GFP was attached to a streptavidin-coated nanobead through a biotinylated anti-GFP nanobody (top). Dynamic MTs were grown from stable MT seeds on coverslips in the presence of Dam1C and Ndc80C-GFP–coated nanobeads (Ndc80C nanobeads) and observed by TIRF microscopy (bottom). **(D)** Images in time sequence (top) show that an Ndc80C nanobead formed MT lateral attachment (0–16 s time points) and end-on attachment (24–68 s) in the presence of Dam1C. Time 0 s was set arbitrarily. Scale

bar, 5 µm. Also refer to diagrams (right). Kymograph was obtained for the same MT (bottom). Scale bars, 5 µm (horizontal) and 60 s (vertical). **(E)** Images in time sequence (top) show that an Ndc80C nanobead formed a MT lateral attachment (0–8 s) and an end-on attachment (14–20 s) and subsequently detached from the MT end (end-on drop-off; 22 s), in the absence of Dam1C. After the nanobead had detached from the MT end, it was not visible by TIRF microscopy, because it was no longer close to the coverslip. Also refer to diagrams (right). Kymograph was obtained for the same MT (bottom). Scale bars are as in D. **(F)** Graph shows the percentage of events (rescue at nanobead, end-on drop-off, and continuous end-on attachment) observed in the absence (n = 54) and presence (n = 101) of Dam1C. The difference between the two groups is significant (****, P < 0.0001).

Lampert et al., 2010; Lampert et al., 2013; Sarangapani et al., 2013; Tien et al., 2010), it is unknown whether this interface, formed by the Ndc80C and Dam1C, sufficiently accounts for the differential regulation of lateral and end-on attachments by Aurora B kinase or whether any other factors are required for this regulation. To address these questions, we reconstituted the kinetochore–MT interface in vitro (i.e., in a cell-free system) using recombinant Dam1C and Ndc80C and also with native kinetochore particles (KCps) purified from budding yeast cells.

## Results

### Reconstitution of kinetochore–MT interface in vitro using defined outer kinetochore components

We aimed to reconstitute the kinetochore–MT interface in vitro using nanobeads and recombinant Ndc80C and Dam1C. The diameter of the nanobead was ∼100 nm, which is somewhat larger than the reported size of the inner kinetochore, which is ∼50 nm (Dimitrova et al., 2016; Gonen et al., 2012). Ndc80Cs were attached to the nanobead with the Ndc80C MT-binding domains oriented outward from the bead surface; Ndc80Cs in situ also take this orientation on the inner kinetochore, and their distal ends directly bind MTs (Cheeseman et al., 2006; Ciferri et al., 2008; Wei et al., 2007). The Ndc80C was expressed in, and purified from, insect cells and visualized by its Spc24 component fused with GFP (Ndc80C-GFP; Fig. 1 B).

In addition, Dam1C and Dam1C-GFP (in which the Dad1 component was fused with GFP) were expressed in, and purified from, bacterial cells (Fig. 1 B). Ndc80C-GFP was attached to a streptavidin-coated nanobead through a biotinylated anti-GFP nanobody (Ndc80C nanobead; Fig. 1 C, top). Based on GFP intensity, we estimated that four Ndc80C-GFP molecules, on average, were attached to a single nanobead (Fig. S1 A). This was a slightly fewer than the five or six Ndc80Cs reportedly assembled on the MIND (Mtw1, Nnf1, Nsl1, Dsn1) complexes at a single kinetochore (Dimitrova et al., 2016; Gonen et al., 2012). Dynamic MTs were generated in vitro from guanosine-5′-[(α,β)-methyleno]triphosphate (GMPCPP)–stabilized MT seeds immobilized on coverslips, and observed by total internal reflection fluorescence (TIRF) microscopy (Fig. 1 C, bottom). The Dam1C-GFP was able to track the end of depolymerizing MTs and accumulate there (up to 10- to 30-fold; Fig. S1 B), as reported previously (Asbury et al., 2006; Tanaka et al., 2007; Westermann et al., 2006).

We investigated how Ndc80C (with GFP) nanobeads behave with dynamic MTs and Dam1C (without GFP) in vitro. An Ndc80C nanobead first attached to the lateral side of a MT (lateral attachment; Fig. 1 D). Lateral attachment required Ndc80Cs, since nanobeads without Ndc80C did not show such

lateral attachment (Fig. S1 C). When the laterally attached MT depolymerized and its plus end caught up with the Ndc80C nanobead, the nanobead became tethered at the MT plus end and subsequently tracked this MT end as it continued to depolymerize (end-on attachment; Fig. 1, D and F). In rare cases, the MT subsequently showed regrowth (MT rescue) without forming end-on attachment or Ndc80 nanobeads detached from the MT end following transient end-on attachment (Fig. 1 F). Overall, the Ndc80C nanobeads on dynamic MTs recapitulated in vitro the behavior of an authentic kinetochore in conversion from the lateral to end-on attachment on a single MT in vivo (Tanaka et al., 2007; Tanaka et al., 2005).

In the absence of Dam1C, the end-on attachment could still be formed, but the Ndc80C nanobead often (31% of cases) detached from the MT plus end following transient end-on attachment (Fig. 1, E and F), suggesting that Dam1C stabilizes end-on attachment. This is consistent with the important roles of Dam1C in interactions with Ndc80C in vitro (Lampert et al., 2010; Lampert et al., 2013; Sarangapani et al., 2013; Tien et al., 2010; Volkov et al., 2013) and in formation of end-on attachment of an authentic kinetochore to a MT in vivo (Kalantzaki et al., 2015; Maure et al., 2011; Tanaka et al., 2007). Dam1C also reduced the MT depolymerization rate during the end-on attachment (Fig. S1 D), which is consistent with the general effect of Dam1C on MT depolymerization (Grishchuk et al., 2008; Westermann et al., 2006). The reduced MT depolymerization rate in the presence of Dam1C could also contribute to sustained end-on attachment.

### Direct comparison between end-on and lateral attachments suggests that Dam1 C-terminal phosphorylation by Aurora B alters their relative strengths

The kinetochore–MT error correction relies on differential regulation of kinetochore interaction with the MT lateral side and the MT end (Kalantzaki et al., 2015; Fig. 1 A). However, there has been no assay to directly compare the strengths of the lateral and end-on attachments. Moreover, it is unknown whether the major outer kinetochore components Ndc80C and Dam1C are sufficient to explain such differential regulation. This prompted us to investigate how Ndc80C nanobeads change their associated MTs in vitro. To this end, we observed situations where two MTs cross each other, one of which has an end-on attachment to an Ndc80C nanobead during depolymerization (MT crossing assay; Fig. 2 A). This assay has two possible outcomes: the end-on attachment may continue and the Ndc80C nanobead passes across the other MT (Fig. 2 A, blue rectangle); alternatively, the Ndc80 nanobead may be transferred from the end of the original MT to the lateral side of the other MT as the depolymerizing MT crosses it (Fig. 2 A, orange rectangle).

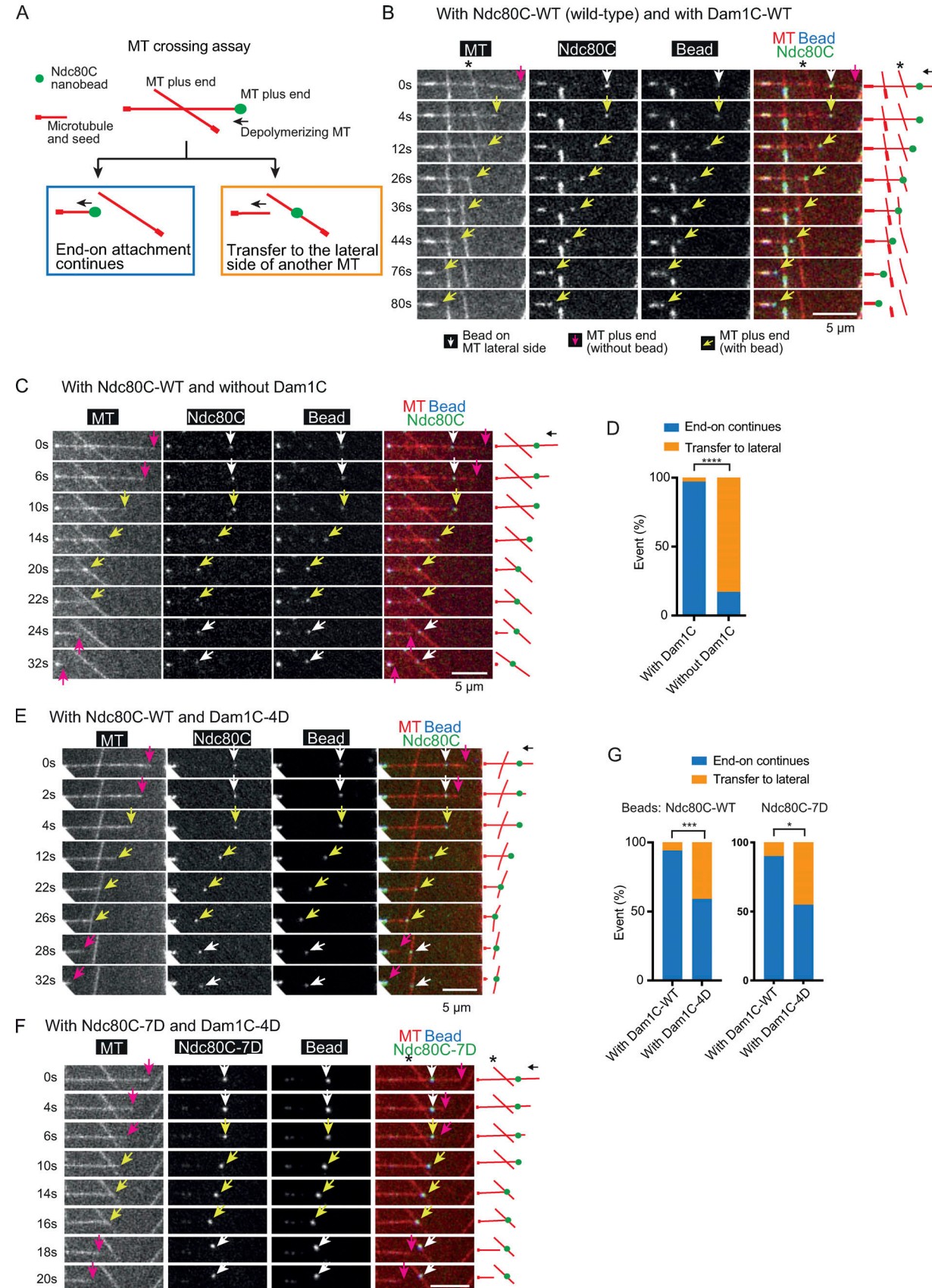

Figure 2. **Direct comparison of strength between the MT lateral and end-on attachments to an Ndc80C nanobead. (A)** Diagram explains the MT crossing assay. Two MTs cross each other, and one forms end-on attachment to the Ndc80C nanobead and depolymerizes. Two possible outcomes are shown:

(i) end-on attachment continues (blue rectangle) or (ii) the Ndc80C nanobead is transferred to the side of the other MT (orange rectangle). **(B)** Images in time sequence show that the MT end-on attachment to the Ndc80C (WT)–nanobead continued after it had passed over the lateral side of another MT, in the presence of Dam1C (WT). Time 0 s was set arbitrarily. Scale bar, 5 µm. Also refer to diagrams (right). Asterisk indicates a crossing MT. **(C)** Images in time sequence show that an Ndc80C (WT) nanobead was transferred from the end of one MT to the side of another MT in the absence of Dam1C. Time 0 s was set arbitrarily. Scale bar, 5 µm. Also refer to diagrams (right). Keys are the same as in B. **(D)** Percentage of events in the MT crossing assay in the presence and absence of Dam1C (n = 35 for each): (i) continued end-on attachment of Ndc80C nanobead (blue) and (ii) transfer to the lateral side of another MT (orange). These events are shown in diagram A, inside the rectangles of the same color. Difference between the two groups is significant (****, P < 0.0001). **(E and F)** Images in time sequence show that an Ndc80C (WT) nanobead (E) or Ndc80C-7D–nanobead (F) was transferred from the end of one MT to the lateral side of another MT in the presence of Dam1C-4D. Scale bars, 5 µm. Also refer to diagrams (right). Asterisks indicate a crossing MT. Keys are the same as in B. **(G)** Percentage of events in the MT crossing assay with Ndc80C-WT (WT) nanobeads (left) or Ndc80C-7D nanobeads (right) with Dam1C-WT (WT) or Dam1C-4D (n = 34, 44, 20, and 31 from left to right): (i) continued end-on attachment (blue) and (ii) transfer to the lateral side of another MT (orange). These events are shown in diagram A, inside the rectangles of the same color. The difference between Dam1C-WT or Dam1C-4D is significant in the left and right graphs (***, P = 0.0005; *, P = 0.0125).

We conducted the MT crossing assay in the presence and absence of Dam1C (Fig. 2, B and C). In the presence of Dam1C, the end-on attachment continued in most cases (97%) when the Ndc80C nanobead passed the other MT (Fig. 2, B and D). By contrast, in the absence of Dam1C, in most cases (83%), the Ndc80 nanobead was transferred to the side of the other MT (Fig. 2, C and D). Given that Dam1C accumulates at the end of depolymerizing MTs (Fig. S1 B), this difference can be explained by a weakened end-on attachment in the absence of Dam1C (Fig. 1, E and F). As a result, the affinity to the lateral attachment may surpass that of the end-on attachment, causing transfer of the Ndc80C nanobead to the lateral side of another MT.

The Dam1C is the most important Aurora B substrate for error correction in budding yeast (Cheeseman et al., 2002; Kalantzaki et al., 2015), and its phosphorylation sites are clustered at the C terminus of Dam1 protein, a component of the Dam1C. It is thought that error correction continues while Dam1 is phosphorylated and stops when biorientation is established and Dam1 is dephosphorylated (Keating et al., 2009). In addition to Dam1 phosphorylation, phosphorylation of the N terminus of Ndc80 protein (a component of the Ndc80C) by Aurora B also contributes to error correction (Akiyoshi et al., 2009). To investigate how Dam1 and Ndc80 phosphorylation by Aurora B affects kinetochore–MT interaction in vitro, we expressed (1) Dam1C carrying four phosphomimic mutations at the C terminus of Dam1 in bacteria and (2) Ndc80C carrying seven phosphomimic mutations at the N terminus of Ndc80 in insect cells. The purified mutant Dam1Cs were called Dam1C-4D-GFP and Dam1C-4D, with and without GFP fusion to Dad1, respectively (Fig. S2 A). The mutant Ndc80Cs purified from insect cells were called Ndc80C-7D-GFP (with GFP fused to Spc24; Fig. S2 A). Dam1C-4D-GFP tracked the plus end of a depolymerizing MT and accumulated there (Fig. S2, B and C), as did Dam1C-WT-GFP (Fig. S1 B). Crucially, Dam1C-4D was able to support continuous end-on attachment of Ndc80C-WT nanobeads in most cases without causing their detachment from the MT ends, as was Dam1C-WT (Fig. S2 D). Meanwhile, when Ndc80C-7D-GFP was attached to a nanobead (Ndc80C-7D nanobead), the nanobead showed lateral and end-on attachments to dynamic MTs, similarly to Ndc80C-WT nanobeads (Fig. 1 D).

Subsequently, we used Dam1C-4D and Ndc80-7D nanobeads in the MT crossing assay (Fig. 2 A). In the presence of Dam1C-WT, the Ndc80C-WT and Ndc80C-7D nanobeads continued end-on attachment to a shrinking MT after crossing another MT in

94% and 90% cases, respectively (Fig. 2, B and G). In the presence of Dam1C-4D, Ndc80C-WT and Ndc80C-7D nanobeads were directly transferred from the plus end of one MT to the lateral side of another MT in 41% and 45% cases, respectively (Fig. 2, E–G). We measured the angle between the two MTs when an Ndc80C-WT nanobead was transferred from the end of one MT to the lateral side of another in the presence of Dam1C-4D. Transfer occurred when two MTs crossed at a wide variety of angles ranging from 27° to 152° (Fig. S2 E). Similarly, Ndc80-7D nanobead transfer occurred at a wide variety of angles.

These results suggest that the Dam1 phosphorylation by Aurora B kinase plays a key role in changing the relative strengths of the end-on and lateral attachments. That is, end-on attachment is stronger in the absence of Dam1 phosphorylation, but it often becomes weaker than lateral attachment when Dam1 is phosphorylated. Our data suggest that Ndc80 phosphorylation may not make a major contribution to this process, which is consistent with the relatively minor roles of Ndc80 phosphorylation in error correction in yeast cells (Akiyoshi et al., 2009; Kalantzaki et al., 2015). Together with our previous observation in yeast cells (Kalantzaki et al., 2015), we suggest that the end-on attachment is specifically weakened by Dam1 phosphorylation by Aurora B, while the lateral attachment strength is unchanged, resulting in alteration in relative strengths of end-on and lateral attachments. This alteration likely drives the exchange of kinetochore–MT interactions (i.e., from end-on attachment on one MT to the lateral attachment on another MT) during error correction. Our in vitro system includes only the Ndc80C and Dam1C components of the kinetochore, thus showing that these two components sufficiently account for the differentially regulated end-on and lateral attachments during error correction.

### Evidence that phosphorylation of the Dam1 C terminus by Aurora B kinase disrupts interaction between Dam1C and Ndc80C during error correction

How could the Dam1 C-terminal phosphomimic mutants promote transfer of an Ndc80C nanobead from the end of one MT to the lateral side of another in the MT crossing assay (Fig. 2, E–G)? The Dam1 C terminus physically interacts with Ndc80C and its phosphorylation (or phosphomimic mutants) weakens this interaction (Kalantzaki et al., 2015; Kim et al., 2017; Lampert et al., 2010; Sarangapani et al., 2013; Tien et al., 2010). However, it has been reported that the Dam1 C terminus also interacts with MTs, and that its phosphorylation (or phosphomimic mutants)

weakens this interaction too (Legal et al., 2016; Zelter et al., 2015). Therefore, phosphomimic mutants at the Dam1 C terminus could disrupt either Dam1C–Ndc80C interaction or Dam1C–MT interaction when an Ndc80C nanobead is detached from the MT end and is transferred to the lateral side of another MT.

To identify the point of disruption, we attached His-tagged Ndc80Cs (WT, without GFP; Fig. 3 A) to nanobeads (which are visible with fluorescence) using a biotinylated anti-His antibody (Fig. 1 C) and compared the behavior of Dam1C-WT-GFP and Dam1C-4D-GFP (Figs. 1 B and S2 A) in the MT crossing assay (Fig. 3 B). The end-on attachment continued in most cases with Dam1C-WT-GFP (Fig. 3, B and C), as with Dam1C (WT, without GFP; Fig. 2, B and D). Moreover, Ndc80C nanobeads were often transferred to the lateral side of another MT with Dam1C-4D-GFP (Fig. 3, B and D), as with Dam1C-4D (without GFP; Fig. 2, E–G). We observed the location of Dam1C-4D-GFP shortly after this transfer occurred (Fig. 3, B and E). If the Dam1C–Ndc80C interaction was disrupted by Dam1 phosphomimic mutants, then Dam1C-4D-GFP signals would continue to track the end of a depolymerizing MT, thus moving away from the Ndc80C nanobead that has been transferred to the side of another MT (Fig. 3 E, left). Alternatively, if the Dam1C–MT interaction was disrupted by Dam1 phosphomimic mutants, then the Dam1C-4D-GFP would remain on the Ndc80C nanobead after it is transferred to the lateral side of another MT (Fig. 3 E, right).

In all the cases where Ndc80C nanobeads were transferred to the side of another MT, Dam1C-4D-GFP signals continued tracking the end of a depolymerizing MT, moving away from the Ndc80C nanobead (Fig. 3, D and E). We could not detect any Dam1C-4D-GFP signals on Ndc80C nanobeads after their transfer to the MT lateral side. For comparison, we looked at the location of Ndc80C-GFP signals soon after transfer of a nanobead to the side of another MT in the presence of Dam1C-4D (without GFP; Fig. 2 E); in all these cases, Ndc80C-GFP signals remained on nanobeads after this transfer (Fig. 3 E). Thus, it can be ruled out that Dam1C-4D was present on nanobeads in a comparable (or larger) amount of Ndc80C after the nanobead transfer. We conclude that phosphomimic mutants at the Dam1 C terminus disrupt the Dam1C–Ndc80C interaction, rather than the Dam1C–MT interaction, to promote transfer of an Ndc80C nanobead from the end of one MT to the lateral side of another. This suggests that phosphorylation of the Dam1 C terminus by Aurora B kinase disrupts the end-on attachment specifically by weakening interaction between Dam1C and Ndc80C during kinetochore–MT error correction.

## Kinetochore particles (KCps) do not show direct transfer between two MTs in the presence of Dam1 and Ndc80 phosphomimic mutants

The Ndc80C nanobead system enabled direct comparison between the strengths of the end-on and lateral attachments. Intriguingly, when Ndc80C nanobeads were transferred from the end of one MT to the lateral side of another (with Dam1-4D phosphomimic mutants), the end-on attachment was not lost until the lateral attachment was formed (i.e., the Ndc80C nanobeads were always attached to one or two MTs during the

transfer; Fig. 2, E and F). In other words, they were "directly" transferred between MTs. Such direct transfer possibly reflects the behavior of native kinetochores during error correction. Indeed, Nicklas and colleagues implied that erroneous MT attachments are not released from the kinetochore until a new attachment is formed in grasshopper spermatocytes (Nicklas, 1997; Nicklas and Ward, 1994).

To address how native yeast kinetochores are transferred between MTs, we purified native KCps from budding yeast, using a FLAG tag fused to Dsn1 (a component of the kinetochore MIND complex) for immunoprecipitation (Akiyoshi et al., 2010). The KCp contained either WT Ndc80 (Ndc80-WT) or phosphomimic mutant Ndc80-7D (at seven Aurora B phosphorylation sites in the N terminus of Ndc80) with three copies of GFP at the Ndc80 C terminus. Before purification of the KCp, we depleted Dam1 protein in yeast cells in which an auxin-inducible degron (AID) tag (Nishimura et al., 2009) was fused to the C terminus of Dam1 by treatment with auxin (1-naphthaleneacetic acid [NAA]). The Western blots of the total cell lysates confirmed depletion of most Dam1 protein after NAA treatment (Fig. 4 Ai). As previously reported (Akiyoshi et al., 2010), the purified KCp contained a wide range of kinetochore components, including both inner and outer kinetochore components, but not Aurora B kinase (Fig. 4 Aii and Table S1).

We reconstituted KCp–MT interactions in vitro by mixing the purified KCp (with Ndc80-WT or -7D) and recombinant Dam1C (-WT or -4D) with dynamic MTs generated in vitro. The KCp first attached to the lateral side (lateral attachment) of dynamic MTs. Then, when the plus end of a depolymerizing MT caught up with the KCp, the KCp attached to the MT end and continuously tracked the end of shrinking MTs (end-on attachment) in the majority of cases (58–70%; Fig. S3, A and D). In other cases (29–40%), the MT subsequently showed regrowth (MT rescue) from the KCp without forming the end-on attachment (Fig. S3, B and D). The continuous end-on attachment relied on the Dam1C, since its absence led to frequent (34–44%) detachment of KCp from the end of shrinking MTs (end-on drop-off) following transient end-on attachment (Fig. S3, C and D). We reason that Ndc80C–Dam1C interactions support sustained end-on attachment (see the section above); in addition, the reduced MT depolymerization rate by Dam1C (Fig. S3 E) may also contribute to sustained end-on attachment. With Dam1C-WT and Dam1C-4D, results were similar, and end-on drop-off of the KCp was rarely found (Fig. S3 D). KCps with Ndc80-WT and Ndc80-7D showed similar results (Fig. S3 D). Thus, the phosphomimic mutants of Dam1 (Dam1-4D) and Ndc80 (Ndc80-7D in KCp) did not significantly affect sustained end-on attachment of the KCp; this is similar to their effects in the Ndc80C nanobeads system (Fig. S2 D).

When the plus end of a depolymerizing MT caught up with the laterally attached Ndc80C nanobead or the purified KCp, MT rescue happened more frequently at the KCp (Fig. S3 D) than at the Ndc80C nanobead (Figs. 1 F and S2 D). In budding yeast, we previously found that similar MT rescue at the kinetochore is facilitated in vivo by Stu2, an orthologue of vertebrate XMAP215/ch-TOG, which localizes at the kinetochore (Gandhi et al., 2011). More frequent MT rescue at the KCp than at the

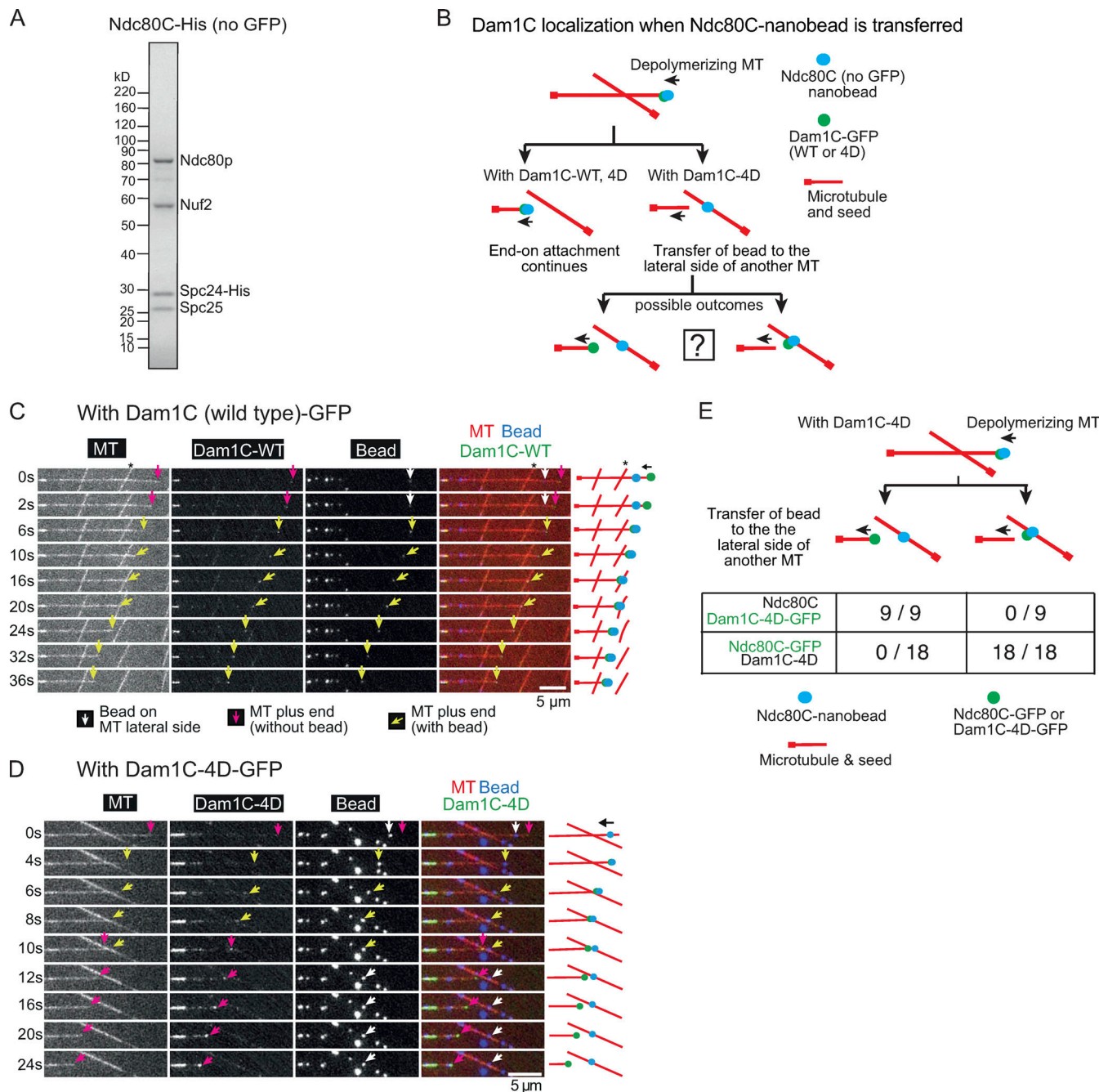

Figure 3. **Dam1C-4D disrupts the interaction with Ndc80C during the transfer of an Ndc80C nanobead between MTs. (A)** Coomassie Blue–stained gel showing purified Ndc80C-His (without GFP and with His tag at the C terminus of Spc24). **(B)** Diagram explains various outcomes in the MT crossing assay regarding the position of an Ndc80C nanobead and the location of Dam1C-GFP (WT or 4D) signals. **(C)** Images in time sequence show that the MT end-on attachment to an Ndc80C (WT) nanobead continued after it had passed across the side of another MT in the presence of Dam1C (WT)-GFP. Dam1C (WT)-GFP signals were at the end of the depolymerizing MT with nanobead throughout the end-on attachment. Time 0 s was set arbitrarily. Scale bar, 5 µm. Also refer to diagrams (right). Asterisk indicates a crossing MT. **(D)** Images in time sequence show that an Ndc80C (WT) nanobead was transferred from the end of one MT to the side of another MT in the presence of Dam1C-4D-GFP. Dam1C-4D-GFP signals tracked the end of depolymerizing MT, moving away from the Ndc80C nanobead. Time 0 s was set arbitrarily. Scale bar, 5 µm. Also refer to diagrams (right). Keys are the same as in C. **(E)** Frequency of events in the MT crossing assay. The cases where Ndc80C nanobeads were transferred to the lateral side of another MT, with Ndc80C (WT, no GFP) and Dam1C-4D-GFP or with Ndc80C (WT)-GFP and Dam1C-4D (no GFP), were investigated. After transfer of the nanobead to the lateral side of another MT, Dam1C-4D-GFP always tracked the end of the original MT (9 out of 9 events), while Ndc80C-GFP always located at the nanobead (18 out of 18 events). The difference between Dam1C-4D-GFP and Ndc80C (WT)-GFP is significant (P < 0.0001).

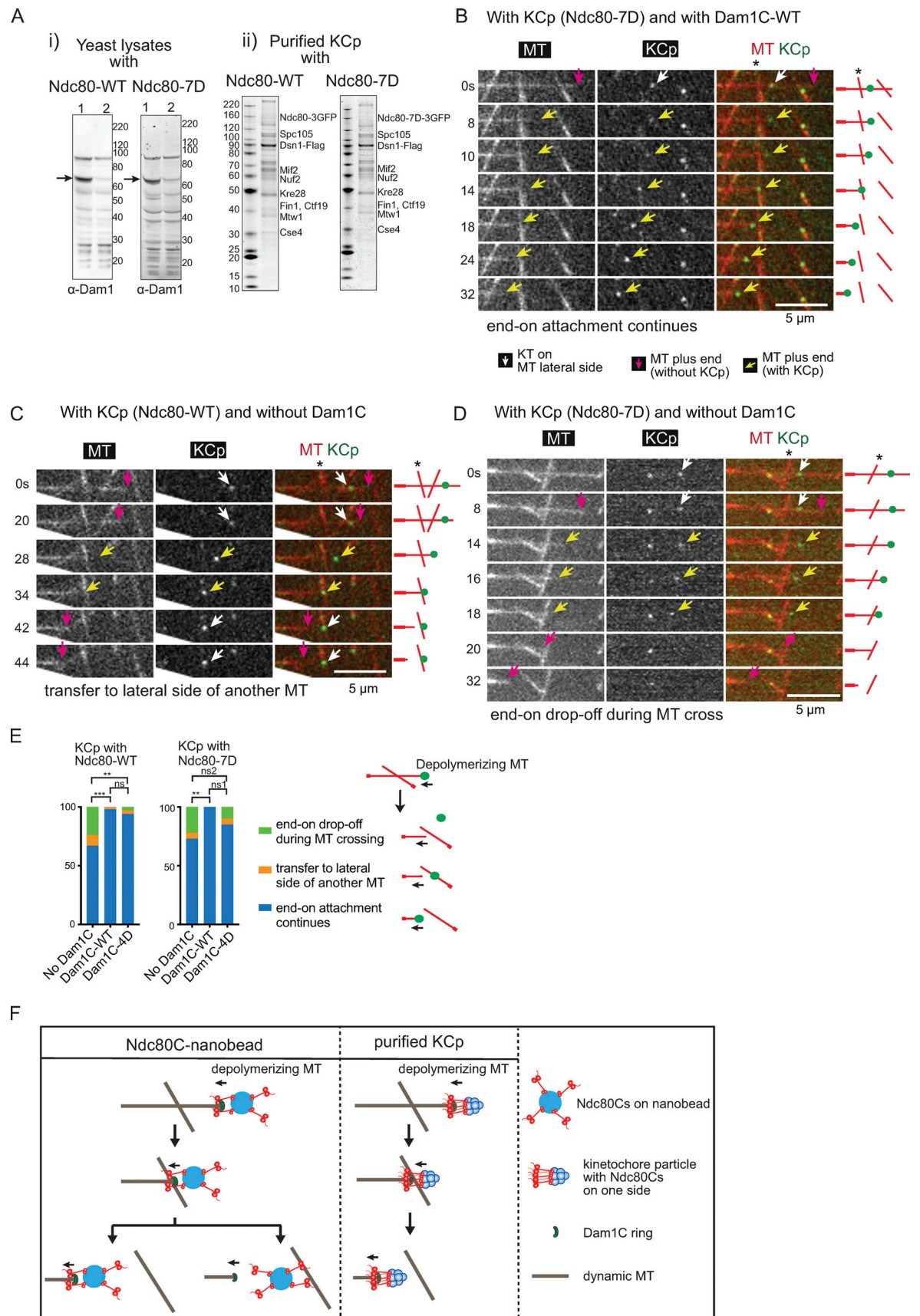

Figure 4. **Behaviors of purified kinetochore particles (KCps) on dynamic MTs in vitro. (A)** i: Western blots of yeast cell lysates from *dam1-aid* strains with *NDC80*-WT (WT; left) and -7D (right). Cells were harvested before (lane 1) and after (lane 2) NAA treatment. The blots were probed with anti-Dam1 antibody.

Arrows indicate Dam1-AID protein. Right: protein size markers (kDa). ii: Purified KCp from *dam1-aid* strains with *NDC80* -WT (left) and -7D (right). Proteins were separated by SDS-PAGE and stained with SyproRuby. Proteins were identified by mass spectrometry (Table S1) and are shown at predicted sizes. Left: protein size markers (kDa). **(B)** Images in time sequence in a MT crossing assay shows that the MT end-on attachment to the KCp (purified from Dam1-depleted cells) continued after it had passed over the lateral side of another MT, in the presence of recombinant Dam1C. Also refer to diagrams (right). Asterisk indicates a crossing MT. Scale bar, 5 µm. In this example, KCp contained Ndc80-7D, and recombinant Dam1C-WT was added to system. **(C)** Images in time sequence show that the KCp (with Ndc80-WT, purified from Dam1-depleted cells) was transferred from the end of one MT to the lateral side of another MT (∼34 s) in the absence of recombinant Dam1C. Keys are the same as in B. **(D)** Images in time sequence show that the KCp (with Ndc80-7D, purified from Dam1-depleted cells) detached from the end of a shrinking MT while passing over the lateral side of another MT (between 18 and 20 s) in the absence of recombinant Dam1C. After the KCp had detached from the MT end, it was not visible by TIRF microscopy, because it was no longer close to the coverslip. Keys are the same as in B. **(E)** Percentage of various outcomes in the MT crossing assay for the KCp with Ndc80-WT (left) or Ndc80-7D (right). The KCp was purified from Dam1-depleted cells. The MT crossing assay was conducted in the absence of recombinant Dam1C or in the presence of recombinant Dam1C-WT or Dam1C-4D (from left to right: $n$ = 45, 44, and 36 for the KCp with Ndc80-WT and $n$ = 37, 30, and 39 for the KCp with Ndc80-7D). Outcomes include (1) continued end-on attachment (blue), (2) transfer to the lateral side of another MT (orange), or (3) end-on drop-off during crossing of another MT (green). Also refer to the diagram (right). Comparisons between two Dam1 conditions give the following: Ndc80-WT, ***, $P$ = 0.0005; ns, $P$ = 0.53; **, $P$ = 0.0086; Ndc80-7D, **, $P$ = 0.0085; ns1, $P$ = 0.080' ns2, $P$ = 0.39. **(F)** Diagram shows the behaviors of Ndc80C nanobeads and the purified KCp in the MT crossing assay. The Ndc80C nanobead was often transferred to the lateral side of another MT in the presence of Dam1C-4D. By contrast, most KCps continued to track the end of a shrinking MT while passing over the other MT in the presence of Dam1C-4D. This difference may be explained by the difference in distribution and orientation of Ndc80Cs. The Ndc80Cs are randomly distributed around the 100-nm nanobeads and orient in all directions, whereas Ndc80Cs on the KCp may have a smaller footprint and orient mostly in one direction toward a MT (Dimitrova et al., 2016; Gonen et al., 2012).

Ndc80C nanobead is explained by Stu2 being present in the purified KCp (Table S1), but not in the Ndc80C nanobead. Our previous in vivo study also showed that some *dam1* mutants (*dam1-1* and *dam1* with C terminus deletion), but not Dam1-WT, showed "end-on standstill" (i.e., an MT neither polymerized nor depolymerized with the kinetochore tethered at its plus end; Kalantzaki et al., 2015; Tanaka et al., 2007); such end-on standstills were not observed with Dam1C-WT or Dam1C-4D or in the absence of Dam1C in the current in vitro study using the Ndc80C nanobead or purified KCp.

Next, we conducted the MT crossing assay with the KCp, as with Ndc80C nanobeads. The KCp contained Ndc80-WT or 7D, and recombinant Dam1C was either present or absent in the system; where present, it contained either Dam1-WT or Dam1-4D. The assay gave three different outcomes: (1) end-on attachment of KCp continued and passed a crossing MT (Fig. 4 B), (2) the KCp was transferred from the end of the original MT to the lateral side of another MT as the depolymerizing MT crossed it (Fig. 4 C), or (3) the KCp detached from the MT end but was not transferred to another MT when the KCp reached it (end-on drop-off; Fig. 4 D). Fig. 4 E shows frequency of these outcomes in blue, orange, and green bars, respectively.

In the MT crossing assay with KCp (with Ndc80-WT), the absence of Dam1C led to more frequent transfer to the lateral side of another MT (Fig. 4, C and E; orange, 9%) and end-on drop-offs (Fig. 4 E; green, 24%) in comparison with the presence of Dam1C-WT. Also, KCp with Ndc80-7D showed similar outcomes in the absence of Dam1C (Fig. 4, D and E). In the presence of recombinant Dam1C-WT and -4D, end-on attachment continued in most cases (Fig. 4, B and E), and the other two events were rare. The outcomes were similar irrespective of Ndc80-WT or -7D in the KCp (Fig. 4 E). In summary, phosphomimic mutants of either Dam1 or Ndc80 showed no significant effects on the behavior of the KCp in the MT crossing assay. With Dam1C-4D, on rare occasions, the KCp was transferred from the end of one MT to the lateral side of another, in contrast to the behavior of Ndc80C nanobeads.

We addressed if this difference in behavior of the Ndc80C nanobeads and the KCp is due to their difference in affinity to the MT lateral side. They did not show significant difference in their affinity to the MT lateral side (Fig. S3 F). We speculate that their different behavior in the MT crossing assay is due to different distributions and orientations of the Ndc80Cs; that is, Ndc80Cs are randomly distributed around the 100-nm nanobeads and orient in all directions, whereas Ndc80Cs on the KCp may have a smaller footprint and orient mostly in one direction (toward a MT; Dimitrova et al., 2016; Gonen et al., 2012; Fig. 4 F). The results with purified KCp suggest that direct transfer between MTs, without generating completely unattached kinetochores, may not be a feature of authentic kinetochores during error correction, at least in budding yeast.

## Discussion

During the process of establishing chromosome biorientation, aberrant interactions could be formed between kinetochores and MTs. These interactions must be detected, dissolved, and reformed in a process called error correction, in which Aurora B kinase plays central roles (Hauf et al., 2003; Lampson et al., 2004; Tanaka et al., 2002). Study of the end-on attachment of kinetochores to MTs using optical tweezers has demonstrated that end-on attachment is weakened by the action of Aurora B (Sarangapani et al., 2013). We previously showed that while Aurora B weakens the end-on attachment, the lateral attachment is impervious to Aurora B regulation (i.e., the end-on and lateral attachments are differentially regulated; Kalantzaki et al., 2015). This led us to propose the model that, during error correction, an end-on attachment is disrupted by the action of Aurora B and then replaced by lateral attachment to a different MT, which is then converted to end-on attachment (Kalantzaki et al., 2015; Fig. 1 A).

However, for this model to work, Aurora B needs to change the relative strengths of the end-on and lateral attachments. Here, we directly compared the strength of end-on and lateral attachments by reconstituting a kinetochore–MT interface in vitro using Ndc80C nanobeads and Dam1C-loaded dynamic MTs. Our results suggest that without Aurora B activity end-on attachment is stronger than lateral attachment, but this relative

strength is reversed when Dam1C is phosphorylated by Aurora B. Because the Ndc80C nanobead assay uses only recombinant Ndc80C and Dam1C components of the kinetochore, our results demonstrate that these two components are sufficient for the differential regulation of end-on and lateral attachments by Aurora B.

Our study also revealed mechanisms by which Dam1C phosphorylation by Aurora B changes the relative strength between end-on and lateral attachments. The Dam1C accumulates at the MT end and interacts with Ndc80C, thus forming end-on attachment (Kalantzaki et al., 2015; Tanaka et al., 2007; Westermann et al., 2006). By visualizing Dam1C in the Ndc80C nanobead assay, we suggested that the Dam1C–Ndc80C interaction is the major disruption point when Dam1 C terminus is phosphorylated by Aurora B. In this way, Aurora B activity specifically weakens the end-on attachment (weaker than the lateral attachment) and promotes the exchange to the lateral attachment to another MT.

When Ndc80C nanobeads were transferred from the MT end to the lateral side of another MT in the presence of Dam1 phosphomimic mutants, this transfer was direct (i.e., the end-on attachment was not lost until the lateral attachment was formed). Does this reflect the behavior of authentic kinetochores during error correction? In fact, it was previously implied that erroneous MT attachments are not released from the kinetochore until a new attachment is formed in grasshopper spermatocytes (Nicklas, 1997; Nicklas and Ward, 1994). However, purified KCps from yeast cells rarely showed such a direct transfer between MTs, in contrast to the Ndc80C nanobeads, suggesting that it may not be a behavior of authentic kinetochores, at least in budding yeast. We suspect that the difference between the Ndc80C nanobeads and the KCps is due to difference in distribution and orientation of the Ndc80Cs (Fig. 4 F). However, it is also possible that additional regulators (e.g., phosphatases counteracting phosphorylation of Dam1C and Ndc80C) are involved in the purified KCps, and consequent regulations may prevent their direct transfer from the MT end to the side of another MT.

If error correction of kinetochore–MT interactions does not involve direct transfer of kinetochores between MTs, then disruption of end-on attachment may occur before a lateral attachment to another MT is formed. However, Dam1 phosphomimic mutants rarely showed detachment of either Ndc80C nanobeads or purified KCps from a MT end without a crossing MT, even if the end-on attachment was weakened (Figs. S2 D and S3 D). Moreover, Dam1 phosphomimic mutants showed only slow (over 30 min or longer) kinetochore detachment from the MT end in cells (Kalantzaki et al., 2015). To explain these observations, we speculate that rapid disruption of the end-on attachment may happen only in the context of aberrant kinetochore–MT interactions such as syntelic attachment, where both sister kinetochores interact with MTs from the same spindle pole. For example, if two MTs from the same spindle pole, attached to sister kinetochores, have different dynamics, then the resulting twisting force would disrupt the weakened end-on attachment of one of sister kinetochores (Ault and Rieder, 1992). Once this happens, the twisting force

would be released and the end-on attachment to the other sister kinetochore would not be disrupted for the time being, even if it is weakened. Thus, such a mechanism would prevent both sister kinetochores simultaneously losing MT attachment and therefore would avoid a chromosome drifting away from the spindle during error correction.

Although the present work using purified KCps suggests that the direct transfer is not the feature of authentic kinetochores in budding yeast, the direct transfer model cannot be ruled out completely. For example, phosphomimic Dam1 and Ndc80 mutants may not weaken the end-on attachment of the purified KCp as efficiently as authentic phosphorylation, thus preventing direct transfer of KCp in this study. Alternatively, although we used porcine tubulins in our study, as they are readily available, purified KCps may show different behaviors in vitro on reconstituted budding yeast MTs. It will be ideal to use species-matched MTs in future in vitro studies if sufficient amount of functional yeast tubulins can be obtained. Whether the transfer of authentic kinetochores is direct or not, our results from the Ndc80C nanobeads system suggest that Dam1 phosphorylation makes the end-on attachment weaker than the lateral attachment, thus enabling the exchange from the end-on attachment to the lateral attachment to another MT.

In conclusion, our study suggests that Ndc80C and Dam1C are sufficient to constitute key regulation of kinetochore–MT interactions by Aurora B kinase. Dam1 phosphorylation by Aurora B weakens the association with the Ndc80C, disrupts end-on attachment, and promotes the exchange to a new MT lateral attachment. Such exchange of kinetochore–MT interactions promotes error correction to establish biorientation.

## Materials and methods
### Plasmid constructs
To express and purify *Saccharomyces cerevisiae* Ndc80C-GFP, *NDC80*, *GST-NUF2*, *SPC24-GFP*, and *SPC25* were cloned into MultiBac vectors, pFL (*NDC80* and *GST-NUF2*) and pUCDM (*SPC24-GFP* and *SPC25*; Bieniossek et al., 2008). A PreScission cleavage site was inserted between *GST* and *NUF2*. The two plasmids (pT3044 and pT3045, respectively) were then fused by Cre-lox recombination in vitro, making pT3046. After transfection of pT3046 into DH10MultiBac *Escherichia coli* (Bieniossek et al., 2008) bacmid DNA was prepared using the alkaline lysis method. To express and purify Ndc80C-7D-GFP, *NDC80* in pT3044 was replaced with *NDC80-7D* mutant, in which Thr21, Ser37, Thr54, Thr71, Thr74, Ser95, and Ser100 of the *NDC80* N-terminal region were replaced with aspartates (Akiyoshi et al., 2009; Kalantzaki et al., 2015), making pT3352. pT3352 and pT3045 were fused in vitro to make pT3371 expressing Ndc80C-7D-GFP. To express and purify Ndc80C (without GFP), pT3045 was modified by replacing GFP on *SPC24-GFP* with a His tag (*SPC24-His*), making pT3219. pT3044 and pT3219 were fused, making pT3220 with Ndc80C without GFP. The MultiBac system was a gift from Tim Richmond (ETH, Zurich, Switzerland).

To express *S. cerevisiae* Dam1C and Dam1C-GFP components in *E. coli*, plasmid constructs were obtained from Stephen Harrison (Harvard Medical School, Boston, MA) and Trisha Davis

(University of Washington, Seattle, WA; Gestaut et al., 2008; Miranda et al., 2005), respectively. These constructs express Spc34 with His tag at its C terminus. To express Dam1C-4D and Dam1C-4D-GFP components, plasmid constructs were generated by DNA synthesis (DC Biosciences) and cloning to replace four serine residues at C terminus of Dam1 (serines at 257, 265, 292, and 327 positions of Dam1) with aspartates (Cheeseman et al., 2002; Kalantzaki et al., 2015; pT3143 and pT3145, respectively).

## Protein purification

Bacmid DNA with Ndc80C components was transfected into Sf9 insect cells to produce baculovirus. For protein expression, Sf9 cells were grown for 60–72 h at 27°C. The cells were then harvested and washed with PBS (8 mM $Na_2HPO_4$, 2 mM $KH_2PO_4$, 2.7 mM KCl, and 137 mM NaCl, pH 7.4) and stored at –80°C. For purification of Ndc80C proteins, the cells were resuspended in buffer containing 50 mM Hepes, pH 7.4, 300 mM NaCl, 1 mM EDTA, 10% glycerol, 1% Triton X-100, 1 mM DTT, and protease inhibitor cocktail (Roche) and lysed with a Dounce homogenizer and sonication at 4°C. The cell lysate was clarified by centrifugation at 25,000 ×*g* for 45–60 min. The soluble fraction was bound to the GST Sepharose 4B (GE Healthcare) for 90–120 min at 4°C. The unbound fractions were removed on a gravity-flow column and washed with buffer containing 50 mM Hepes, pH 7.4, 250 mM NaCl, 1 mM EDTA, 10% glycerol, and 1 mM DTT. The proteins were eluted in buffer containing 50 mM Hepes, pH 7.4, 150 mM NaCl, 1 mM EDTA, 0.05% Triton X-100, and 1 mM DTT by cleavage with PreScission protease (GE Healthcare), which cleaves between GST and Nuf2 in the GST-Nuf2 fusion protein. The eluted proteins were further purified by gel filtration using Superose 6 Increase 10/300 GL column (GE Healthcare) equilibrated with buffer containing 25 mM Tris-Cl, pH 7.6, 250 mM NaCl, 1 mM EDTA, 3 mM $MgCl_2$, 5% glycerol, and 1 mM DTT. The fractions containing Ndc80C were pooled, and the buffer was exchanged into the one containing 80 mM Pipes, pH 6.8, 1 mM $MgCl_2$, 1 mM EGTA, 150 mM KCl, 5% sucrose, and 0.2 mM DTT using a PD-10 desalting column (GE Healthcare). Purified Ndc80C was stored at –80°C.

For purification of Dam1C proteins, Rosetta 2(DE3) cells (Novagen; 71401) transformed with respective constructs were grown at 37°C until $OD_{600}$ reached 0.6–0.7. Then, protein expression was induced by 0.1 mM IPTG for 22 h at 15–16°C. The cells were harvested and stored at –80°C. Dam1C proteins were purified in a three-step process by affinity, ion-exchange, and gel filtration chromatography, as described previously (Miranda et al., 2005). The bacterial cells were resuspended in ice-cold buffer containing 50 mM sodium phosphate, pH 7.5, 500 mM NaCl, 1 mM EDTA, 0.5% Triton X-100, 20 mM imidazole, 1 mM DTT, and protease inhibitor cocktail (Roche). Cells were then lysed on ice and sonicated at 4°C, and the cell debris was separated by centrifugation at 25,000 ×*g* for 45–60 min. The supernatant was incubated with nickel–nitrilotriacetic acid agarose (Ni-NTA agarose, Qiagen) at 4°C for 90 min. Unbound fractions were separated using a gravity-flow column. Protein-bound nickel–nitrilotriacetic acid agarose was washed with buffer containing 50 mM sodium phosphate, pH 7.5, 500 mM NaCl, 1 mM EDTA, 0.5% Triton X-100, 50 mM imidazole, and

1 mM DTT. Then, proteins were eluted with the same buffer containing 250 mM imidazole, without Triton X-100. The eluted proteins were exchanged into 50 mM sodium phosphate, pH 7.5, 100 mM NaCl, 1 mM EDTA, and 0.2 mM DTT using PD-10 desalting columns (GE Healthcare), and then 1 mM ATP and 250-fold molar excess of synthetic peptide NRLLTG was added and incubated for 1 h at 4°C. Proteins were purified on a MonoQ 5/50 column (GE Healthcare) with a gradient of 100–1,000 mM NaCl. The eluent was mixed again with 1 mM ATP and NRLLTG peptide (250-fold molar excess) and incubated for 1 h at 4°C and purified on a Superose 6 Increase 10/300 GL column (GE Healthcare) equilibrated with 25 mM sodium phosphate, pH 7.4, 500 mM NaCl, 1 mM EDTA, and 0.2 mM DTT. The fractions corresponding to Dam1C proteins were pooled and the buffer was exchanged into the one containing 80 mM Pipes, pH 6.8, 1 mM $MgCl_2$, 1 mM EGTA, 150 mM KCl, 5% sucrose, and 0.2 mM DTT. Purified Dam1C was stored at –80°C.

## Attaching Ndc80C to nanobeads

Fluorescent (excitation at 647 nm) streptavidin-coated magnetic nanobeads (100 nm diameter; will be referred as nanobeads hereafter) were obtained from Creative Diagnostics (WHM-ME647). Ndc80C-GFP, Ndc80C-7D-GFP, and Ndc80C-His were attached to nanobeads using biotinylated anti-GFP nanobody (Chromotek; gtb-250) and biotinylated anti-His antibody (Qiagen; 34440), as appropriate, as shown in Fig. 1 C (top). For this, the beads were incubated with biotinylated anti-GFP nanobody (or biotinylated anti-His antibody) for ~1 h along with 5 mg/ml BSA at 4°C. The unbound fractions were removed after the nanobeads were bound to the magnet and washed three times in MRB80 buffer (80 mM Pipes, pH 6.8, 1 mM $MgCl_2$, and 1 mM EGTA) supplemented with 5 mg/ml BSA. The nanobeads were then incubated with Ndc80C in MRB80 buffer supplemented with 5 mg/ml BSA for 1 h at 4°C, washed three times, and finally resuspended in MRB80 buffer.

## Generation of dynamic MTs on coverslips

Purified tubulin proteins were obtained from Cytoskeleton. For preparation of MT seeds, 20 µM porcine tubulin mix containing 18% biotinylated tubulin, 12% rhodamine-labeled tubulin, and 70% unlabeled tubulin was incubated with 1 mM GMPCPP (Jena BioScience; NU-405S) on ice and subsequently at 37°C for 30 min. MTs were separated from free tubulin by centrifugation using an Airfuge (Beckman Coulter) for 5 min. The MTs were subjected to another round of depolymerization and polymerization with 1 mM GMPCPP, and the final MT seed samples were stored in MRB80 buffer (80 mM Pipes, pH 6.8, 1 mM $MgCl_2$, and 1 mM EGTA) supplemented with 10% glycerol.

Coverslips were plasma cleaned using Carbon Coater (Agar Scientific) and treated with PlusOne Repel-Silane (GE Healthcare) for 10–15 min. The coverslips were further cleaned by sonication in methanol and finally rinsed in water. Flow chambers were assembled with cleaned coverslips and microscopy slides using double-sided tape.

The chambers were treated with 0.2 mg/ml PLL-PEG-biotin (Surface Solutions) in MRB80 buffer for 5 min. Subsequently, they were washed with buffer and incubated with 1 mg/ml

NeutrAvidin (Thermo Fisher Scientific) for 5 min. MT seeds were attached to the coverslips with the biotin-NeutrAvidin links and then incubated with NeutrAvidin once again to neutralize the exposed biotins on MT seeds that were already bound to coverslips. Finally, the chambers were incubated with 1 mg/ml κ-casein.

The in vitro reaction mixture was prepared in MRB80 buffer (80 mM Pipes, pH 6.8, 1 mM $MgCl_2$, and 1 mM EGTA) consisting of 12 μM tubulin mix (11.5 μM unlabeled tubulin and 0.5 μM rhodamine tubulin), 50–60 mM KCl, 2 mM $MgCl_2$, 0.1% methylcellulose, 0.5 mg/ml κ-casein, 1 mM GTP, 6 mM DTT, oxygen scavenging system (400 μg/ml glucose-oxidase, 200 μg/ml catalase, 4 mM DTT, and 20 mM glucose), and 10 nM of relevant Dam1C proteins. The mixture was centrifuged for 5 min using Airfuge. To the supernatant, nanobeads coated with Ndc80C-GFP, Ndc80C-7D-GFP, or Ndc80C-His (or purified KCp) were mixed and added to the flow chamber containing MT seeds. The chamber was sealed with vacuum grease and observed at 30°C by TIRF microscopy. To study behavior of Dam1C on dynamic MTs, the reaction mixture was prepared in the same way as above but without nanobeads.

### TIRF microscopy and image analysis

Images of dynamic MTs were acquired by TIRF microscopy using Nikon Eclipse Ti-E (Nikon) inverted research microscope equipped with four diode lasers (405 nm, 488 nm, 561 nm, and 647 nm; Coherent), Acousto-Optic Tunable Filters shutter (Solamere Technology), appropriate filters (Chroma), perfect focus system, the CFI Apochromat TIRF 100× 1.49 N.A. oil objective lens (Nikon) and Evolve Delta electron-multiplying charge-coupled device 512 × 512 camera (Photometrics). The TIRF system was controlled with μ-manager software (Open Imaging; Edelstein et al., 2014). A temperature-control chamber (Okolab) was used to maintain the temperature.

Images were analyzed using ImageJ and OMERO. Kymographs were generated in time sequence along a chosen line for an individual MT using the KymoResliceWide plugin on ImageJ. The Dam1C-GFP intensity at the end of depolymerizing MT was obtained from kymographs along the path of the MT depolymerization using ImageJ. The MT depolymerization rate was determined as the ratio of the length of MT depolymerization (horizontal) to time taken (vertical) in each event on the kymograph.

Affinity of Ndc80C nanobeads and KCps to MTs was analyzed as follows: the average number of Ndc80C(WT) nanobeads and KCps (with Ndc80C-WT) in a given volume was determined using a TIRF microscope (with semi TIRF angle), and then equal numbers of Ndc80C nanobeads and KCps were added to the dynamic MTs to determine their affinity to the MT side.

Statistical analyses were performed with Prism software (GraphPad) using a Fisher's exact test (Fig. 2, D and G; and Fig. 3 E), $\chi^2$ test (Figs. 1 F, 4 E, S2 D, and S3 D), or t test (Fig. S1 D; and Fig. S3, E and F). The fluorescence intensities of GFP protein and that of Ndc80C-GFP on nanobeads were obtained from TIRF microscopy images (in semi-TIRF angle) and analyzed using the ImageJ plugin DoM v.1.1.6 (Detection of Molecules; https://github.com/ekatrukha/DoM_Utrecht). All experiments were repeated at least twice, and similar results were obtained. Results of repeated experiments were combined, and the combined data are shown in figures.

### Yeast strains

Two yeast strains (T12525 and T13763; see their genotypes below) were constructed for purification of KCps (see next section). Three copies of FLAG tags with KAN marker (pT2329) were added to the DSN1 C terminus, three copies of GFPs with KAN marker (Maekawa et al., 2003) were added to the NDC80 C terminus, and AID tag with clonNAT marker (Nishimura et al., 2009) was added to the DAM1 C terminus at their original loci. To do so, these tags and selection markers were amplified by PCR, and yeast cells were transformed with the PCR products. The construct of rice TIR1 under ADH1 promoter was previously reported (Nishimura et al., 2009) and integrated at the TRP1 locus. These constructs were introduced into yeast cells by sequential transformation or gathered in a single yeast strain by crossing strains with each construct. To replace NDC80 (WT) in T12525 with NDC80-7D and make T13763, the 5′ promoter DNA fragment of NDC80, HIS marker, 5′ promoter, and Ndc80-7D ORF were cloned in this order to make the replacement cassette pT2037. Genotypes of yeast strains are as follows: T12525 MATα dsn1::DSN1-3XFLAG::KAN-MX4 ndc80::NDC80-3xGFP::KAN dam1::dam1-aid::NAT-NT2 trp1::ADH1p-TIR1-9Myc::TRP1; T13763 MATα dsn1::DSN1-3XFLAG::KAN-MX4 ndc80::HIS3::Ndc80-7D-3xGFP::KAN dam1::dam1-aid::NAT-NT2 trp1::ADH1p-TIR1-9Myc::TRP1.

### Purification of KCps

KCps were affinity purified using FLAG tag on Dsn1 as described previously (Akiyoshi et al., 2010; Gupta et al., 2018) after depletion of Dam1 protein. Briefly, overnight grown yeast cells (T12525 or T13763) were diluted to 0.1 $OD_{600}$ in YPAD (1% yeast extract, 2%peptone, 0.01% adenine hydrochloride, and 2% glucose) medium and grown further at 25°C. When the $OD_{600}$ of the culture reached 0.6, cells were treated with 1 mM NAA for 90 min to deplete Dam1p tagged with auxin-induced degron. The cells were harvested by centrifugation and washed with Milli-Q water and 0.2 mM PMSF and then washed with buffer H (25 mM Hepes, pH 8.0, 150 mM KCl, 2 mM MgCl2, 0.1 mM EDTA, pH 8.0, 0.5 mM EGTA, pH 8.0, 0.1% NP-40, and 15% glycerol) containing protease inhibitors and phosphatase inhibitors. The cells were resuspended in lysis buffer containing buffer H supplemented with protease and phosphatase inhibitors, dropped into liquid nitrogen by pipetting to create "dots," and stored at –80°C. The frozen "dots" were ground into powder in a freezer mill for 2 min and cooled down for 2 min; this cycle was repeated seven times. The cell powder was thawed on ice and incubated with benzonase (300 U/ml) for 30 min at 4°C. The soluble fraction was separated by centrifugation at 20,000 ×g for 30 min, and the supernatant was centrifuged again at 72,000 ×g for 60 min. The final supernatant was incubated with FLAG antibody conjugated beads (Sigma; anti-FLAG M2 magnetic beads; M8823) for 3 h with constant rotation at 4°C. The beads were washed four times with buffer H containing protease inhibitors, phosphatase inhibitors and 2 mM DTT and washed three more times with buffer H containing protease inhibitors and phosphatase

inhibitors. The bound proteins were eluted in buffer HE (25 mM Hepes, pH 8.0, 150 mM KCl, 2 mM MgCl₂, 0.1 mM EDTA, pH 8.0, 0.5 mM EGTA, pH 8.0, and 15% glycerol) containing protease inhibitors, phosphatase inhibitors and 0.5 mg/ml FLAG peptides (Sigma; F4799-4MG). The eluted proteins were aliquoted and stored at –80°C.

### Western blots
Proteins were separated on an SDS-PAGE gel and transferred to a nitrocellulose membrane using iBlot 2 apparatus (Thermo Fisher Scientific). The membrane was incubated in TBST (20 mM Tris, 150 mM NaCl, and 0.1% Tween 20) containing 5% milk for 1 h, washed three times with TBST, and incubated with the primary antibody—the polyclonal anti-Dam1p (Keating et al., 2009) or monoclonal anti-GFP antibody (Sigma; 11814460001) overnight at 4°C. The blots were incubated with either HRP-tagged secondary antibody (Sigma; A3415-.5ML; Cell Signaling Technologies; 7076P2) or IRDye secondary antibody (LI-COR, 926–68074 and 926–32212). The blots were then scanned using either with ChemiDoc (Bio-Rad) or LI-COR odyssey imager.

### Mass spectrometry
KCps were purified as described above and eluted in buffer containing 0.2% RapiGest SF surfactant (Waters) and 50 mM Hepes, pH8.0. The protein sample (~150 ng) was digested in solution with trypsin overnight at 30°C and processed with HiPPR Detergent Removal Spin Column Kit (Thermo Fisher Scientific; 88305), and the sample was dried by SpeedVac at room temperature and stored at –20°C.

Liquid chromatography–mass spectrometry analysis was done at the FingerPrints Proteomics Facility (University of Dundee). Analysis of peptide readout was performed on a Q Exactive plus mass spectrometer (Thermo Fisher Scientific) coupled with a Dionex Ultimate 3000 RS (Thermo Fisher Scientific). Liquid chromatography buffers used were buffer A (0.1% formic acid in Milli-Q water) and buffer B (80% acetonitrile and 0.1% formic acid in Milli-Q water). Samples were resuspended in 35 µl of 1% formic acid, and 15 µl was loaded onto a trap column (Thermo Fisher Scientific; PepMap nanoViper C18 column, 5 µm, 100 Å) equilibrated in 0.1% trifluoroacetic acid (TFA). The trap column was washed for 5 min at the same flow rate with 0.1% TFA and then switched in-line with a Thermo Fisher Scientific resolving C18 column (PepMap RSLC C18 column, 2 µm, 100 Å). The peptides were eluted from the column at a constant flow rate of 300 nl/min with a linear gradient from 2% buffer B to 5%, then to 35%, and finally to 98%. The column was then washed with 98% buffer B for 20 min and reequilibrated in 2% buffer B for 17 min. The column was kept at a constant temperature of 50°C. Q-exactive plus was operated in data-dependent positive ionization mode. The source voltage was set to 2.5 kV, and the capillary temperature was 250°C. A scan cycle comprised MS1 scan (m/z range from 350 to 1,600, ion injection time of 20 ms, resolution 70,000, and automatic gain control $10^6$) acquired in profile mode, followed by 15 sequential-dependent MS2 scans (resolution 17,500) of the most intense ions fulfilling predefined selection criteria. The

higher-energy collisional dissociation collision energy was set to 27% of the normalized collision energy. Mass accuracy was checked before the start of samples analysis. For protein identification, MaxQuant version 1.6.0.16 (Tyanova et al., 2015) was run against *S. cerevisiae* protein database.

### Online supplemental material
Fig. S1 (associated with Fig. 1) provides additional data about Ndc80C nanobeads (A), Dam1C on dynamic MTs in vitro (B), and interactions of Ndc80C nanobeads with dynamic MTs in vitro (C and D). Fig. S2 (associated with Fig. 2) shows additional data about phospho-mimic mutants of Ndc80 and Dam1 (A–D) and from the MT crossing assay (E). Fig. S3 (associated with Fig. 4) provides additional data about purified KCps on dynamic MTs in vitro (A–F). Table S1 shows the mass spectrometry analysis of purified KCps.

## Acknowledgments
We thank Tanaka laboratory members, E.R. Griffis, E.A. Katrukha, R. Cross, D. Peet, T. Surrey, G. Ball, M. Gierlinski, and M. Miller for discussion; S. Biggins (Fred Hutchinson Cancer Research Center, Seattle, WA) for the KCp purification protocol; S. Harrison (Harvard Medical School, Boston, MA), T. Davis (University of Washington, Seattle, WA), T. Richmond (ETH, Zurich, Switzerland), L. Garcia (University of Dundee, Dundee, UK), and Addgene for reagents; E.R. Griffis, P. Appleton, S. Swift, D. Bajer, and Dundee Imaging Facility for TIRF microscope setting and maintenance; Dundee FingerPrints Proteomic Facility for mass spectrometry analyses; N. Sheidaei for help in protein purification; and the OMERO team for technical help.

This work was supported by the Wellcome Trust (grants 096535/Z/11/Z, 097945/B/11/Z, and 219418/Z/19/Z) and the Medical Research Council (grant K015869).

The authors declare no competing financial interests.

Author contributions: H. Doodhi: conceptualization, formal analysis, investigation, visualization, methodology, and writing (original draft and review and editing). T. Kasciukovic: resources and methodology. L. Clayton: resources, methodology, and writing (review and editing). T.U. Tanaka: conceptualization, supervision, visualization, methodology, writing (original draft and review and editing), project administration, and funding acquisition.

Submitted: 20 November 2020

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

# Supplemental material

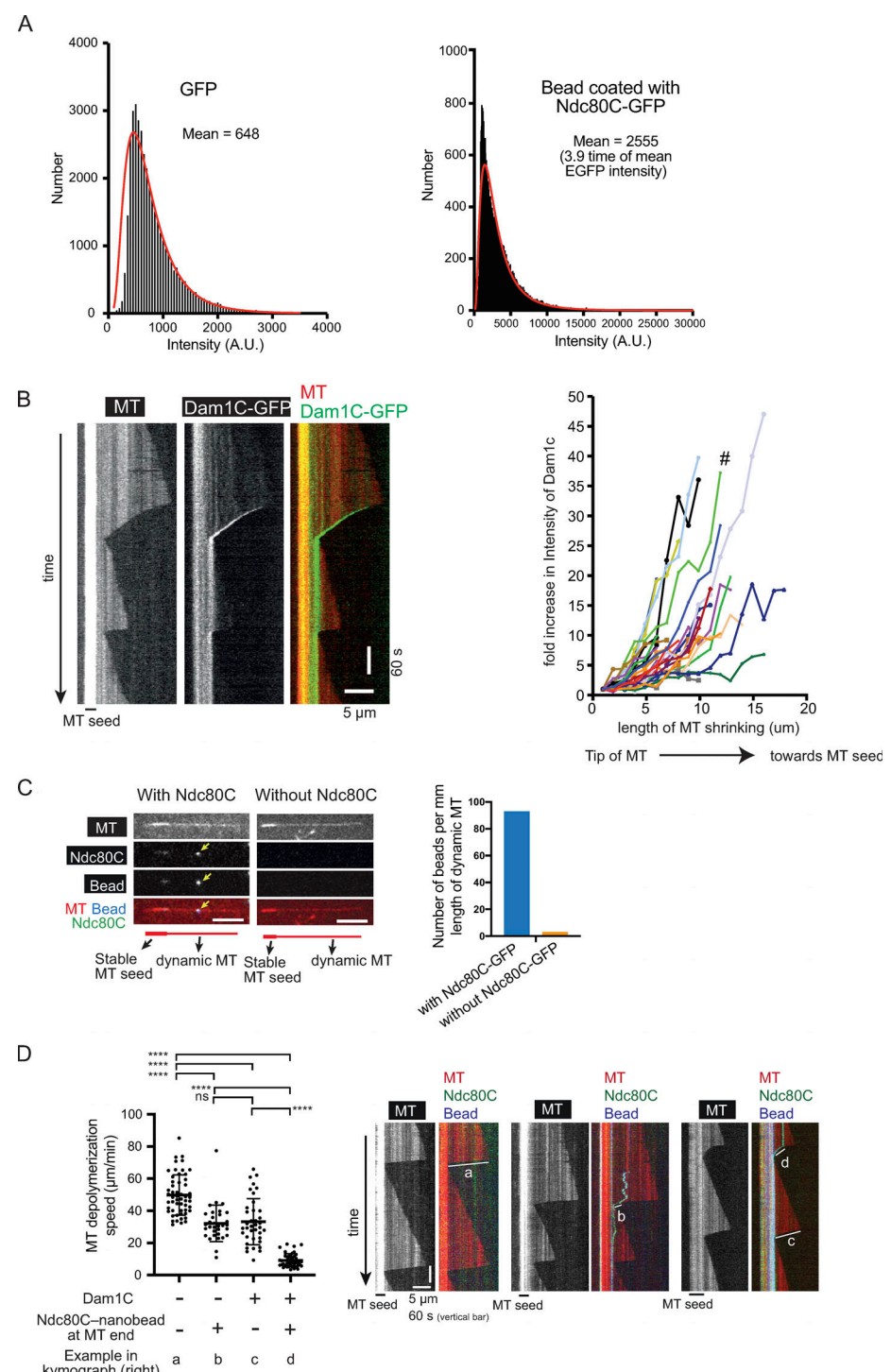

Figure S1. **Supplemental data associated with Fig. 1. (A)** Signal quantification of GFP (by itself, not fused to Ndc80C) and Ndc80C-GFP–coated beads. Black bars (bin size 50) represent the number of fluorescent spots with the indicated intensity. The red line is the lognormal fit of the quantification. A.U., arbitrary unit. **(B)** Kymograph (left) shows that the Dam1C-GFP signal tracked the end of a depolymerizing MT. Scale bars, 5 μm (horizontal) and 60 s (vertical). Graph (right) shows fold increases of Dam1C-GFP signals at the plus ends of individual MTs. The Dam1C-GFP signal for the first 1 μm after their appearance was averaged and set to one for normalization. The Dam1C signal during subsequent MT shrinkage was averaged in 1-μm increments, normalized, and plotted against the length of MT shrinkage. # shows the fold increase of the example shown in the kymograph (left). **(C)** Images show a representative example of dynamic MTs when they were incubated with Ndc80C-GFP–coated nanobeads (left) or control nanobeads without Ndc80C-GFP (right). The yellow arrowhead indicates an Ndc80C-GFP–coated nanobead on the lateral side of a dynamic MT. Scale bar, 5 μm. Graph shows the number of nanobeads (with and without Ndc80C-GFP coating) on dynamic MTs per millimeter length of MTs. **(D)** Graph shows MT depolymerization rates either in the absence or presence of Dam1C in solution and in the absence or presence of Ndc80C nanobead at the end of depolymerizing MT (n = 54, 38, 32, and 44 from left to right). Error bars represent standard deviation. The MT depolymerization rates determined from the kymographs as shown on the right. Scale bars, 5 μm (horizontal) and 60 s (vertical). **** P < 0.0001; ns, not significant (P = 0.73).

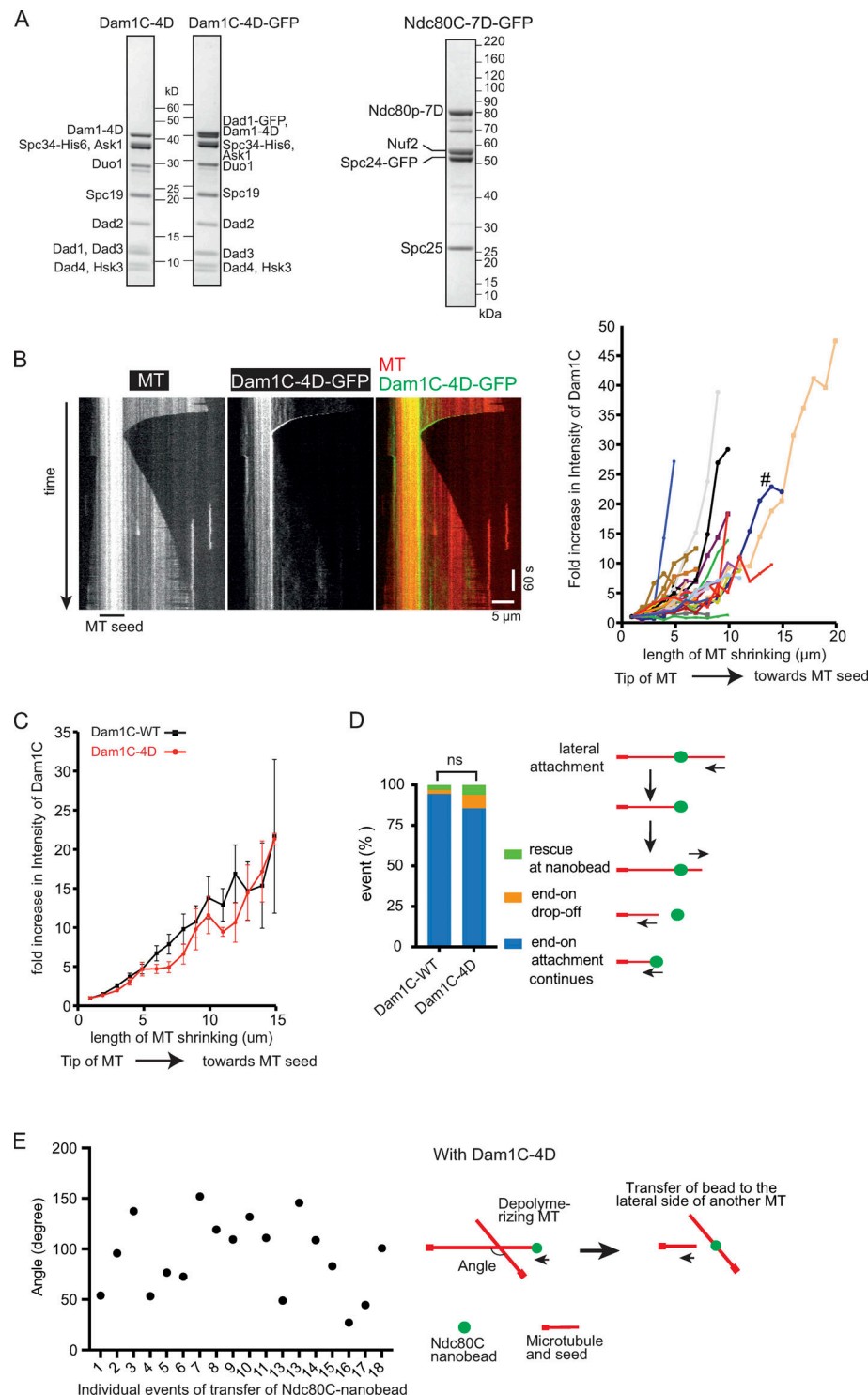

Figure S2. **Supplemental data associated with Fig. 2. (A)** Coomassie Blue stained gel (left panel) shows purified Dam1C-4D (in which four serine residues at the C terminus of Dam1 were replaced with aspartate) with and without GFP at the C terminus of Dad1. Coomassie Blue stained gel (right panel) shows purified Ndc80C-7D-GFP (with seven serine residues at N terminus of Ndc80 replaced with aspartate). **(B)** Kymograph (left) shows that Dam1C-4D-GFP signals tracked the end of a depolymerizing MT. Scale bar, 5 µm (horizontal) and 60 s (vertical). Graph (right) shows fold increases of Dam1C-4D-GFP signals at the plus ends of individual MTs, which were obtained and plotted as in Fig. S1 B. # shows the fold increase in the example shown in the kymograph (left). **(C)** Dam1C (WT)-GFP and Dam1C-4D-GFP show similar fold increases at the plus end of shrinking MTs. The fold increase of Dam1C (WT)-GFP ($n$ = 28; black squares) or Dam1C-4D-GFP ($n$ = 32; red circles) signals at the shrinking MT ends (Figs. S1 B and S2 B) was averaged among multiple MTs and plotted against the length of MT shrinkage. Error bars show SEM. **(D)** Graph shows the percentage of events; rescue at Ndc80C nanobead, end-on drop-off and continuous end-on attachment, observed in the presence of Dam1C WT (WT; $n$ = 126) and Dam1C-4D ($n$ = 97). The difference between the two groups is not significant (ns, P = 0.07). **(E)** Angles made by two MTs between which Ndc80C (WT) nanobeads were transferred, from end-on to the lateral side of another MT, in the presence of Dam1C-4D. Angles were measured in individual events as shown in diagram (right) and plotted (left).

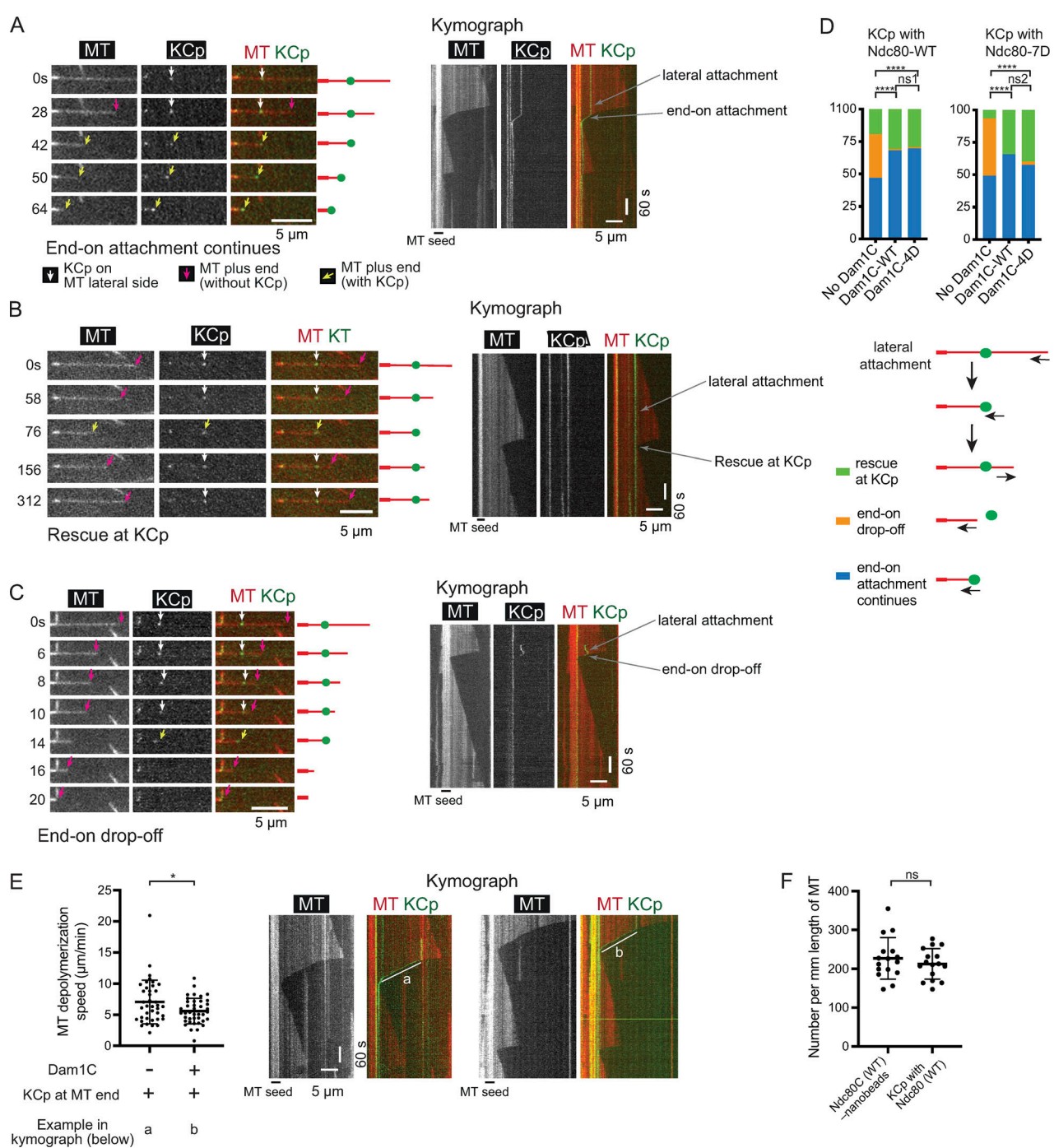

Figure S3. **Supplemental data associated with Fig. 4. (A)** Images in time sequence show that the KCp (purified from Dam1-depleted cells) is attached to the lateral side of a MT (white arrows, 0–28 s) and formed the end-on attachment (yellow arrows, 42–64 s) in the presence of recombinant Dam1C. Kymograph obtained for the same MT is shown on the right side. In this example, the KCp contained Ndc80C-WT, and recombinant Dam1C-WT was added to the system. **(B)** Images in time sequence show that the KCp (purified from Dam1-depleted cells) is attached to the lateral side of a MT (white arrows, 0–58 s), shrinking MT reaches KCp (yellow arrows, 76 s), and MT regrows (magenta arrows, 76–312 s). Kymograph obtained for the same MT is shown on the right side. In this example, the KCp contained Ndc80-WT and recombinant Dam1C-4D was added to the system. **(C)** Images in time sequence show that the KCp (purified from Dam1-depleted cells) is attached to the lateral side of a MT (white arrows, 0–10 s), formed the end-on attachment (yellow arrows, 14 s), and detached from the MT end (14–16 s) in the absence of recombinant Dam1C. After the KCp had detached from the MT end, it was not visible by TIRF microscopy, because it was no longer close to the coverslip. Kymograph obtained for the same MT is shown on the right side. In this example, the KCp contained Ndc80-7D. **(D)** Percentage of events (rescue at KCp [green], end-on drop-off [orange], and continuous end-on attachment (blue), observed for the KCp with Ndc80-WT (left) or Ndc80-7D [right]) soon after the end of shrinking MTs reached the KCp. The KCp was purified from Dam1-depleted cells. Experiments were conducted in the absence of recombinant Dam1C or in the presence of recombinant Dam1C-WT or Dam1C-4D (from left to right: n = 98, 91, and 99 for the KCp with Ndc80-WT; n = 61, 76, and 80 for the KCp with Ndc80-7D). ****, P < 0.0001; ns1, P = 0.97; ns2, P = 0.26. **(E)** MT depolymerization rates in the absence (left, n = 39) and presence (right, n = 40) of recombinant Dam1C-WT, when KCp (Nd80-WT) was tracking the end of depolymerizing MTs. Difference between the two groups is significant

(P = 0.03). Error bars represent SD. Kymographs (right) show representative events of KCp tracking the end of a depolymerizing MT (white lines a and b). Scale bars, 5 µm (horizontal) and 60 s (vertical). **(F)** Number of Ndc80C (WT) nanobeads or KCp (with Ndc80-WT) that bind the lateral side of dynamic MTs in each microscope field of view (normalized to per millimeter length of MT) after equal numbers of Ndc80C nanobeads and KCp were added to the system. Difference between the two groups is not significant. ns, P = 0.40. Error bars represent SD.

**Table S1 is provided online and lists kinetochore proteins identified in purified KCps by mass spectrometry.**

