## [Peer Review File · The Journal of Cell Biology]

Aurora B switches relative strength of kinetochore-microtubule attachment modes for error correction

Harinath Doodhi, Taciana Kasciukovic, Lesley Clayton, and Tomoyuki Tanaka

Corresponding Author(s): Tomoyuki Tanaka, University of Dundee and Harinath Doodhi, University of Dundee

Review Timeline:

Submission Date:	2020-11-20
Editorial Decision:	2020-12-23
Revision Received:	2021-03-03
Editorial Decision:	2021-03-10
Revision Received:	2021-03-14

Monitoring Editor: Arshad Desai

Scientific Editor: Melina Casadio

Transaction Report:

DOI: <https://doi.org/10.1083/jcb.202011117>

December 23, 2020

Re: JCB manuscript #202011117

Prof. Tomoyuki U Tanaka
University of Dundee
School of Life Sciences
Wellcome Trust Biocentre
Dow Street
Dundee DD1 5EH
United Kingdom

Dear Tomo,

Thank you for submitting your manuscript entitled "Aurora B switches relative strength of kinetochore-microtubule attachment modes to promote error correction". The manuscript was assessed by expert reviewers, whose comments are appended to this letter. Based on their feedback, we are interested in receiving a revised version that addresses the major comments.

In preparing your revision, please note the comments from reviewers on clarifying specific points and streamlining the text. The reviewers appreciated the clear delineation of the difference observed between the reconstituted system and the kinetochore particles and, while experimental resolution of this difference is not necessary in the revision, the description and interpretation needs to be improved following their suggestions. Reviewer #3 raises two points about the mechanism that should be addressable by analyzing microtubule dynamics parameters in your current data. Reviewer #2 notes that species-matched tubulin is not assessed - this caveat should be acknowledged in the text but experimental analysis with yeast tubulin is not necessary for the revision.

GENERAL GUIDELINES:

Text limits: Character count for an Article is < 40,000, not including spaces. Count includes title page, abstract, introduction, results, discussion, acknowledgments, and figure legends. Count does not include materials and methods, references, tables, or supplemental legends.

Figures: Articles may have up to 10 main text figures. Figures must be prepared according to the policies outlined in our Instructions to Authors, under Data Presentation, <https://jcb.rupress.org/site/misc/ifora.xhtml>. All figures in accepted manuscripts will be screened prior to publication.

*****IMPORTANT:** It is JCB policy that if requested, original data images must be made available. Failure to provide original images upon request will result in unavoidable delays in publication. Please ensure that you have access to all original microscopy and blot data images before

submitting your revision.***

Supplemental information: There are strict limits on the allowable amount of supplemental data. Articles may have up to 5 supplemental figures. Up to 10 supplemental videos or flash animations are allowed. A summary of all supplemental material should appear at the end of the Materials and methods section.

As you may know, the typical timeframe for revisions is three to four months. However, we at JCB realize that the implementation of social distancing and shelter in place measures that limit spread of COVID-19 also pose challenges to scientific researchers. Lab closures especially are preventing scientists from conducting experiments to further their research. Therefore, JCB has waived the revision time limit. We recommend that you reach out to the editors once your lab has reopened to decide on an appropriate time frame for resubmission. Please note that papers are generally considered through only one revision cycle, so any revised manuscript will likely be either accepted or rejected.

Thank you for this interesting contribution to Journal of Cell Biology. You can contact us at the journal office with any questions, cellbio@rockefeller.edu or call (212) 327-8588.

Sincerely,

Arshad Desai, PhD
Editor, Journal of Cell Biology

Melina Casadio, PhD
Senior Scientific Editor, Journal of Cell Biology

Reviewer #1 (Comments to the Authors (Required)):

The Aurora B kinase plays crucial roles in correction of erroneous kinetochore-microtubule attachments. Although it is well established that phosphorylation of microtubule-binding kinetochore proteins by Aurora B leads to weakened binding in vitro, it still remains unclear how error correction actually works in vivo. Based on their previous finding in budding yeast that Aurora B weakens end-on attachment but not lateral attachment, the authors' group proposed that, during error correction, an end-on attachment is disrupted by Aurora B and subsequently replaced by lateral attachment to a different microtubule. In this manuscript by Doodhi and colleagues, they established an elegant microtubule crossing assay to directly compare the strengths of lateral v.s. end-on attachments. They show that the phosphomimetic Dam1-4D mutant efficiently promotes exchange of end-on attached Ndc80-coated beads to lateral attachment on a different microtubule and that it does so by weakening its interaction with Ndc80. Based on these results, (I think) they propose that Aurora B promotes error correction by direct exchange of end-on attached kinetochores to lateral attachment of a different microtubule. In contrast, native kinetochore

particles containing Dam1-4D fail to show such exchanges and the authors conclude that direct transfer between microtubules may not be a feature of authentic kinetochores during error correction.

Overall, experiments are carefully performed and controlled. This manuscript describes the first in vitro assay to directly compare the attachment strength of end-on v.s. lateral attachment and provides a potentially important conceptual advance, both of which are important contributions to the field. I therefore support publication of this manuscript in *Journal of Cell Biology*. However, there are some inconsistencies in the manuscript and it was rather difficult to understand their conclusions.

Comments

- "This alteration likely drives the exchange of kinetochore-MT interactions, i.e. from end-on attachment on one MT to the lateral attachment on another MT, during error correction (page 6)". In my view, this sentence is the most important conceptual advance of this manuscript because current models assume that Aurora B promotes detachment of phosphorylated kinetochores that are bound to microtubules in an end-on manner. However, it is not clear whether the authors are pushing forward this hypothesis. For example, it is not clear from the abstract whether they propose direct transfer (sentence "We suggest that the Dam1 phosphorylation weakens interaction with the Ndc80 complex, disrupts the end-on attachment and promotes the exchange to a new lateral attachment, leading to error correction"). The first paragraph of page 12 is another example. The authors should be clear about their conclusion and claim.

- Although their experiment using native kinetochore particles does not support their claim, they may consider mentioning other potential explanations for it (e.g. presence of other Aurora B targets or inefficiency of phosphomimetic mutants compared to actual phosphorylation) rather than concluding that direct transfer may not be a feature of authentic kinetochores.

- I suggest that the authors remove the last sentence of the abstract because their experiment using native kinetochore particles does not support their hypothesis.

Reviewed by Bungo Akiyoshi

Reviewer #2 (Comments to the Authors (Required)):

In this manuscript, Doodhi, Tanaka and colleagues examine Aurora B-dependent correction of kinetochore-microtubule attachment in an in vitro system using recombinant budding yeast proteins and purified budding yeast kinetochores bound to nanoparticles.

The experiments are technically well performed, and the results are beautifully presented.

The insights from this study are somewhat limited since meticulous live-cell microscopy by the Tanaka lab had already shown the effects of Dam1C-4D and Ndc80-7D on kinetochore-microtubule attachment in vivo (Kalantzaki et al., 2015).

Nevertheless, the in vitro system provides some advances. In particular, being able to observe a relatively large number of events in a purified system provides statistical power and allows conclusions on the relative strength of relevant interactions. From the first part, using recombinant

Ndc80 and Dam1, the authors conclude that phosphorylation of Dam1 by Aurora B (mimicked by a quadruple aspartate mutant) weakens the Aurora B-Dam1 interaction, which favors lateral over end-on attachment. This was largely expected, given prior results.

Figure 4, where the authors coat the beads with purified kinetochores rather than with recombinant Ndc80, adds an interesting twist. Interestingly, the Dam1-4D mutant hardly has an effect in this system. This suggests that there is influence by other factors, which remain to be identified.

Overall, the biggest strength of the paper, in my opinion, lies in establishing this in vitro system, which will now allow testing of additional kinetochore mutants for their in vitro effects.

Major comments:

- The use of porcine tubulin in the in vitro assay is a weakness in my opinion. Tubulin from different species is not necessarily interchangeable (e.g. Howes/Nogales 2017). I am not sure this is technically feasible for the authors, but the conclusions would be more solid if at least some of the key experiments with kinetochore particles had been performed with budding yeast tubulin instead.
- Kinetochore particle-coated and Ndc80-coated nanoparticles behave differently at crossing microtubules when end-on attachment is weak. Whereas Ndc80-coated beads typically transfer to the other microtubule, the kinetochore particle-coated beads do not. The authors attribute this to the different orientation of Ndc80 complexes. However, it is also possible that the purified kinetochore particles generally have a lower affinity for the lateral side of microtubules. Could this be tested?

Minor comments:

- The findings from kinetochore particles are not covered in the abstract. I do not find it absolutely essential to do so, but I think it would allow other researchers to more easily find relevant information.
- Page 5 "The Dam1C is the most important Aurora B substrate for error correction...": Please add "in budding yeast".
- Methods: "All the experiments were repeated at least twice and similar results were obtained.": It remains unclear whether only one of the experiments is shown in the figures or whether the results were combined for the data that is shown in the figures.

Typo:

- Page 8 "confirmed depletion of the most Dam1 protein after the NAA treatment" should be "depletion of most Dam1 protein"

Reviewer #3 (Comments to the Authors (Required)):

This study addresses how Aurora B kinase promotes the correction of aberrant kinetochore-microtubule attachments, which is an essential step for faithful chromosome segregation. It is evolutionary conserved from yeast to human that the phosphorylation of kinetochore proteins by Aurora B kinase is required to release erroneous attachments. However, how exactly Aurora B promotes this process was unclear. Previous work from the same group has shown that Aurora B weakens "end-on" microtubule attachments but not "lateral" microtubule attachments, using budding yeast cells (Kalantzaki, 2015). The authors have now reconstituted this process in vitro with purified kinetochore proteins, Ndc80 complex and Dam1 complex. This in vitro system nicely

recapitulated the dynamic kinetochore-microtubule interactions previously observed in the cell. More interestingly, this system allowed the authors to show that just three factors (i.e., Ndc80 complex, Dam1 complex, and Aurora B) are sufficient to explain why Aurora B kinase weakens end-on attachments but not lateral attachments. The data presented in the manuscript are strong, supporting their major conclusion.

My major comment is on the contribution of Dam1 complex on preventing end-on drop-off of Ndc80-nanobeads (Figure 1D-F) and kinetochore particles (Figure S3). The authors briefly discuss that Dam1's interaction with the Ndc80 complex stabilizes the end-on attachments (page 5, line 2). However, the Dam1-4D mutant, which abolishes this interaction also prevents end-on drop-off of Ndc80-nanobeads (Figure S2D) and kinetochore particles (Figure S3C). Westermann et al. has previously shown that the accumulation of Dam1 complex on microtubule plus-ends slows down microtubule depolymerization (Westermann et al., 2006). This seems to be also true in the kymograph images in this manuscript (Figure S1B, S2B). Therefore, another possible contribution of Dam1 complex in this in vitro system is that it slows down microtubule depolymerization as they accumulate on microtubule tips, which would help Ndc80-nanobeads and kinetochore particles maintain their end-on attachments. I recommend the authors to quantify the microtubule depolymerization rate with and without Dam1 complex (in their existing dataset) to test this possibility.

Another major comment is on the differences between what has been observed during error correction in the cell (Kalantzaki, 2015) and what the authors observed in their in vitro system in the current study. In their previous paper (Kalantzaki, 2015), the authors have followed kinetochore-microtubule dynamics in budding yeast cells and categorized different stages of the error correction process (i.e., end-on pulling (end-on continues in the current study), microtubule rescue, end-on standstill, end-on drop-off). What I realized is that in the current manuscript, two categories are missing (i.e., microtubule rescue, end-on standstill). I would recommend the authors to clarify if these categories are not observed or included in the end-on continues category. This is critical to understand which part of the error-correction process has been reconstituted in this new system. This also applies to the experiments using kinetochore particles (Figure 4 and S3). Regarding the microtubule-rescue category, the authors have previously shown that this process depends on Stu2 (Kalantzaki, 2015). Therefore, it makes sense if this category is not observed in the in vitro system using Ndc80-nanobeads, because Stu2 is not present. However, kinetochore particles that they purified do contain Stu2 (Table S1). So, I would recommend the authors to discuss the different results between in vivo (Kalantzaki, 2015) and this in vitro system. Similarly, the end-on standstill category was the major category of Dam1 mutant cells in multiple studies (Tanaka, 2007 and Kalantzaki, 2015). Therefore, it is important to clarify if this category was not observed in the current study where Dam1 complex was absent or mutated. Discussing the differences between in vivo and in vitro system would lead to a deeper understanding of the in vitro system that the authors have developed and make the manuscript even stronger.

Additional comment:

This reviewer felt that the Discussion section is somewhat redundant with the Result section, and there are not so much new insights/future direction provided except for the second last paragraph. For example, authors could develop more on (1) different results between the in vivo study (Kalantzaki, 2015) and the in vitro system (as mentioned above), (2) different results between Ndc80-nanobeads and kinetochore particles, and (3) why Ndc80-7D has little impact on error correction in budding yeast compared to other model organisms based on the results from this in vitro system.

Response to Reviewers

We would like to thank the Reviewers for their insightful comments. We have attempted to address the points raised by them, as below.

In the following sentences, *Reviewers' comments* are shown in *italic (specific points we address are underlined)*. Our response is shown in plain text following '>'.

Reviewer #1

"This alteration likely drives the exchange of kinetochore-MT interactions, i.e. from end-on attachment on one MT to the lateral attachment on another MT, during error correction (page 6)". However, it is not clear whether the authors are pushing forward this hypothesis. For example, it is not clear from the abstract whether they propose direct transfer. The first paragraph of page 12 is another example. The authors should be clear about their conclusion and claim.

> The end-on attachment was not lost until the lateral attachment was formed, when Ndc80C–nanobeads were transferred from the microtubule (MT) end to the lateral side of another MT in the MT crossing assay, in the presence of Dam1-4D i.e. the transfer was direct. However, purified kinetochore particles did not show such behaviour. From these results, we think that the direct transfer may not reflect the behaviour of authentic kinetochores in budding yeast cells. Nonetheless, the Ndc80C–nanobead system is a useful tool to directly compare strength of the end-on and lateral attachments.

> Based on the result with purified kinetochore particles, we reason that a new lateral attachment is formed after the authentic kinetochore detaches from the MT end (indirect transfer). But we speculate that this may happen only in context of syntelic attachment, as described in Discussion (5th paragraph). However, we do not have experimental evidence substantiating this speculation. Therefore, although we prefer indirect transfer, we would rather not make a strong conclusion on whether the transfer is direct or indirect. The two sentences on pages 6 and 12, indicated by the reviewer, are meant to be neutral regarding whether the transfer is direct or indirect. Whether the transfer of authentic kinetochores is direct or indirect, our results from the Ndc80C–nanobeads system support that Dam1 phosphorylation makes the end-on attachment weaker than the lateral attachment, thus enabling the exchange from the end-on attachment to the lateral attachment to another MT – for clarification, we added this sentence to the Discussion (6th paragraph).

Although their experiment using native kinetochore particles does not support their claim, they may consider mentioning other potential explanations for it (e.g. presence of other Aurora B targets or inefficiency of phosphomimetic mutants compared to actual phosphorylation) rather than concluding that direct transfer may not be a feature of authentic kinetochores.

> Responding to this request, we have now included the following sentence in Discussion (4th paragraph): it is also possible that additional regulators (e.g. phosphatases counteracting phosphorylation of Dam1C and Ndc80C) are involved in the purified kinetochore particles and consequent regulations may prevent their direct transfer from the MT end to the side of another MT.

> Although we prefer that the direct transfer is not a feature of authentic kinetochores based on our results of purified native kinetochores, we cannot completely rule out that the direct transfer happens to authentic kinetochores. For example, as the reviewer indicates, phospho-mimic Dam1 and Ndc80 mutants may not weaken the end-on attachment of the

purified kinetochore particle as efficiently as authentic phosphorylation, which may prevent the direct transfer. We have mentioned such possibility in Discussion (6th paragraph).

I suggest that the authors remove the last sentence of the abstract because their experiment using native kinetochore particles does not support their hypothesis.

> As discussed above, the last sentence in the abstract 'We suggest that the Dam1 phosphorylation weakens interaction with the Ndc80 complex, disrupts the end-on attachment and promotes the exchange to a new lateral attachment, leading to error correction' does not specify whether disruption of the end-on attachment occurs before (indirect transfer) or after (direct transfer) the formation of new lateral attachment. Therefore, the sentence is still valid whether the transfer is direct or indirect.

Reviewer #2

- The use of porcine tubulin in the in vitro assay is a weakness in my opinion. Tubulin from different species is not necessarily interchangeable (e.g. Howes/Nogales 2017). I am not sure this is technically feasible for the authors, but the conclusions would be more solid if at least some of the key experiments with kinetochore particles had been performed with budding yeast tubulin instead.

> It is technically difficult to purify functional tubulin proteins from *S. cerevisiae* cells in large quantity sufficient for our *in vitro* analyses, at least in our hands. Responding to the reviewer's criticism, we have included the following sentences: 'although we used porcine tubulins in our study as they are readily available, purified kinetochore particles may show different behaviours *in vitro* on reconstituted budding-yeast MTs. It will be ideal to use species-matched MTs in future studies if sufficient amount of functional yeast tubulins can be obtained.' (6th paragraph in Discussion).

- Kinetochore particle-coated and Ndc80-coated nanoparticles behave differently at crossing microtubules when end-on attachment is weak. Whereas Ndc80-coated beads typically transfer to the other microtubule, the kinetochore particle-coated beads do not. The authors attribute this to the different orientation of Ndc80 complexes. However, it is also possible that the purified kinetochore particles generally have a lower affinity for the lateral side of microtubules. Could this be tested?

> To address the underlined point raised by the reviewer, we have compared the affinity of Ndc80C–nanobeads and purified kinetochore particles to the microtubule lateral side. When the two kinds of particles were used in the same concentration and their numbers on microtubules (per unit length) were counted and compared, there was no significant difference between the two (Figure S3F), suggesting that Ndc80C–nanobeads and purified kinetochore particles show similar affinity to the microtubule lateral side.

- The findings from kinetochore particles are not covered in the abstract. I do not find it absolutely essential to do so, but I think it would allow other researchers to more easily find relevant information.

> Major conclusions in this study have been obtained from the Ndc80C–nanobeads system. Nonetheless we agree with the reviewer that it helps readers find relevant information, if we briefly mention use of purified kinetochore particles in abstract. Thus, we have added a short sentence 'Similar reconstitutions with purified kinetochore particles were used for comparison' to abstract. Because of a strict word limitation of abstract, we

could not include more information on purified kinetochore particles in abstract.

- Page 5 "The Dam1C is the most important Aurora B substrate for error correction...":
Please add "in budding yeast".

> The requested phrase has been added.

- Methods: "All the experiments were repeated at least twice and similar results were obtained.": It remains unclear whether only one of the experiments is shown in the figures or whether the results were combined for the data that is shown in the figure.

> The results of repeated experiments were combined and the combined data are shown in Figures. We clarified this point in Method.

- Page 8 "confirmed depletion of the most Dam1 protein after the NAA treatment" should be "depletion of most Dam1 protein"

> This has been amended as suggested.

Reviewer #3

My major comment is on the contribution of Dam1 complex on preventing end-on drop-off of Ndc80-nanobeads (Figure 1D-F) and kinetochore particles (Figure S3). another possible contribution of Dam1 complex in this in vitro system is that it slows down microtubule depolymerization as they accumulate on microtubule tips, which would help Ndc80-nanobeads and kinetochore particles maintain their end-on attachments. I recommend the authors to quantify the microtubule depolymerization rate with and without Dam1 complex (in their existing dataset) to test this possibility.

> Following this suggestion, we have compared the microtubule (MT) depolymerization rate during end-on attachment of the Ndc80C–nanobead, in the presence and absence of the Dam1C (Figure S1D). The MT depolymerization rate was indeed lower in the presence of Dam1C than in its absence. We have also obtained a similar result in comparison of the MT depolymerization rate during end-on attachment of the purified kinetochore particle in the presence and absence of the Dam1C (Figure S3E). Thus, we agree with the reviewer that slower MT depolymerization with the Dam1C may contribute to sustained end-on attachment of the Ndc80–nanobeads and kinetochore particles. We have included these results and discussions in the Results section (Page 5 and 9).

Another major comment is on the differences between what has been observed during error correction in the cell (Kalantzaki, 2015) and what the authors observed in their in vitro system in the current study. In their previous paper (Kalantzaki, 2015), the authors have followed kinetochore-microtubule dynamics in budding yeast cells and categorized different stages of the error correction process (i.e., end-on pulling (end-on continues in the current study), microtubule rescue, end-on standstill, end-on drop-off). What I realized is that in the current manuscript, two categories are missing (i.e., microtubule rescue, end-on standstill). I would recommend the authors to clarify if these categories are not observed or included in the end-on continues category. This is critical to understand which part of the error-correction process has been reconstituted in this new system. This also applies to the experiments using kinetochore particles (Figure 4 and S3). Similarly, the end-on standstill category was the major category of Dam1 mutant cells in multiple studies (Tanaka, 2007 and Kalantzaki, 2015). Therefore, it is important to clarify if this category was not observed in the current study where Dam1 complex was absent or mutated....

> Following this suggestion, we analysed MT rescue and end-on standstill soon after the plus end of a depolymerizing MT caught up with the Ndc80C–nanobead or purified KCp on the lateral side of a MT. MT rescue was rare with the Ndc80C–nanobeads in the presence of Dam1C wild-type and Dam1C-4D and without Dam1C (Figure 1F, S2D). On the other hand, MT rescue events were observed at the purified KCp in the presence of Dam1C wild-type/4D (30-40%) and in the absence of Dam1C (7-19%) (Figure S3B, D). As the reviewer indicates, we previously found that such MT rescue is facilitated *in vivo* by Stu2, which localizes at the kinetochore (Gandhi et al 2011). More frequent MT rescue at the KCp than at the Ndc80C–nanobead is explained by Stu2 being present in the purified KCp (Table S1) but not in the Ndc80C–nanobead. We included this argument in the Results section (Page 9-10).

> As the reviewer indicates, our previous *in vivo* study also showed that some *dam1* mutants (*dam1-1* and *dam1* with C-terminus deletion), but not Dam1 wild-type, showed 'end-on standstill' i.e. an MT neither polymerized nor depolymerized with the kinetochore tethered at its plus end (Tanaka K et al, 2007; Kalantzaki et al 2015) – such end-on standstill was not observed with Dam1C wild-type or Dam1C-4D or without Dam1C in the current *in vitro* study using Ndc80C nanobeads or purified KCps. We included this information in the Results section (Page 10).

This reviewer felt that the Discussion section is somewhat redundant with the Result section, and there are not so much new insights/future direction provided except for the second last paragraph. For example, authors could develop more on (1) different results between the in vivo study (Kalantzaki, 2015) and the in vitro system (as mentioned above), (2) different results between Ndc80-nanobeads and kinetochore particles, and (3) why Ndc80-7D has little impact on error correction in budding yeast compared to other model organisms based on the results from this in vitro system.

> To respond to this criticism, we revised several parts of Discussion as follows:

- a) We deleted most parts of paragraphs 3 and 4 in Discussion of the first manuscript to avoid redundancy with the Results section.
- b) We have added a new paragraph to discuss in more detail how Dam1C phosphorylation by Aurora B changes the relative strength between end-on and lateral attachments (3rd paragraph of Discussion in the revised manuscript).
- c) We have added an alternative explanation for different behaviours of Ndc80C–nanobeads and purified kinetochores particles (4th paragraph of Discussion in the revised manuscript).
- d) We have added a new paragraph to discuss alternative mechanisms for the transfer of kinetochores between microtubules during error correction (6th paragraph of Discussion in the revised paragraph).
- e) We have added comparison with our previous *in vivo* studies (Gandhi et al 2011; Kalantzaki et al 2015) (reviewer's point 1) to the Results section (Page 9-10) as it fits better there than in Discussion.

March 10, 2021

RE: JCB Manuscript #202011117R

Prof. Tomoyuki U Tanaka
University of Dundee
School of Life Sciences
Wellcome Trust Biocentre
Dow Street
Dundee DD1 5EH
United Kingdom

Dear Prof. Tanaka,

Thank you for submitting your revised manuscript entitled "Aurora B switches relative strength of kinetochore-microtubule attachment modes for error correction". We have assessed the revision and response to the reviews editorially. We additionally checked in with Reviewer #3, who provided input to us as well. We all find that the revision is improved by the changes made in response to the reviews. The experimental analysis of depolymerization rate is also a nice addition. We would be happy to publish your paper in JCB pending final revisions necessary to meet our formatting guidelines (see details below).

1) eTOC summary: A 40-word summary that describes the context and significance of the findings for a general readership should be included on the title page. The statement should be written in the present tense and refer to the work in the third person.

- Please include a summary statement on the title page of the resubmission. It should start with "First author name(s) et al..." to match our preferred style.

2) Statistical analysis: Error bars on graphic representations of numerical data must be clearly described in the figure legend. The number of independent data points (n) represented in a graph must be indicated in the legend. Statistical methods should be explained in full in the materials and methods. For figures presenting pooled data the statistical measure should be defined in the figure legends.

3) Materials and methods: Should be comprehensive and not simply reference a previous publication for details on how an experiment was performed. Please provide full descriptions in the text for readers who may not have access to referenced manuscripts.

- For all cell lines, vectors, constructs/cDNAs, etc. - all genetic material: please include database / vendor ID (e.g., Addgene, ATCC, etc.) or if unavailable, please briefly describe their basic genetic features *even if described in other published work or gifted to you by other investigators*

- Please include species and source for all antibodies, including secondary, as well as catalog numbers/vendor identifiers if available.

- Sequences should be provided for all oligos: primers, si/shRNA, gRNAs, etc.

- Microscope image acquisition: The following information must be provided about the acquisition

and processing of images:

- a. Make and model of microscope
- b. Type, magnification, and numerical aperture of the objective lenses
- c. Temperature
- d. imaging medium
- e. Fluorochromes
- f. Camera make and model
- g. Acquisition software
- h. Any software used for image processing subsequent to data acquisition. Please include details and types of operations involved (e.g., type of deconvolution, 3D reconstitutions, surface or volume rendering, gamma adjustments, etc.).

4) References: There is no limit to the number of references cited in a manuscript. References should be cited parenthetically in the text by author and year of publication.

- Please abbreviate the names of journals according to PubMed.

5) Author contributions: A separate author contribution section is required following the Acknowledgments in all research manuscripts. All authors should be mentioned and designated by their full names. We encourage use of the CRediT nomenclature.

A. MANUSCRIPT ORGANIZATION AND FORMATTING:

Full guidelines are available on our Instructions for Authors page, <https://jcb.rupress.org/submission-guidelines#revised>. **Submission of a paper that does not conform to JCB guidelines will delay the acceptance of your manuscript.**

B. FINAL FILES:

-- High-resolution figure and video files: See our detailed guidelines for preparing your production-ready images, <https://jcb.rupress.org/fig-vid-guidelines>.

Thank you for this interesting contribution, we look forward to publishing your paper in Journal of Cell Biology.

Sincerely,

Arshad Desai, PhD
Editor, Journal of Cell Biology

Melina Casadio, PhD
Senior Scientific Editor, Journal of Cell Biology